# On Differentially Private Federated Linear Contextual Bandits

**Xingyu Zhou**
Wayne State University, USA
Email: xingyu.zhou@wayne.edu

**Sayak Ray Chowdhury**
Microsoft Research, India
Email: t-sayakr@microsoft.com

## Abstract

We consider cross-silo federated linear contextual bandit (LCB) problem under differential privacy, where multiple silos interact with their respective local users and communicate via a central server to realize collaboration without sacrificing each user's privacy. We identify three issues in the state-of-the-art (Dubey & Pentland, 2020): (i) failure of claimed privacy protection, (ii) incorrect regret bound due to noise miscalculation and (iii) ungrounded communication cost. To resolve these issues, we take a two-step approach. First, we design an algorithmic framework consisting of a generic federated LCB algorithm and flexible privacy protocols. Then, leveraging the proposed framework, we study federated LCBs under two different privacy constraints. We first establish privacy and regret guarantees under silo-level local differential privacy, which fix the issues present in state-of-the-art algorithm. To further improve the regret performance, we next consider shuffle model of differential privacy, under which we show that our algorithm can achieve nearly "optimal" regret without a trusted server. We accomplish this via two different schemes – one relies on a new result on privacy amplification via shuffling for DP mechanisms and another one leverages the integration of a shuffle protocol for vector sum into the tree-based mechanism, both of which might be of independent interest. Finally, we support our theoretical results with numerical evaluations over contextual bandit instances generated from both synthetic and real-life data.

## 1 Introduction

We consider the classic *cross-silo* Federated Learning (FL) paradigm (Kairouz et al., 2021) applied to linear contextual bandits (LCB). In this setting, a set of $M$ local silos or agents (e.g., hospitals) communicate with a central server to learn about the unknown bandit parameter (e.g., hidden vector representing values of the user for different medicines). In particular, at each round $t \in [T]$, each local agent $i \in [M]$ receives a new user (e.g., patient) with context information $c_{t,i} \in \mathcal{C}_i$ (e.g., age, gender, medical history), recommends an action $a_{t,i} \in \mathcal{K}_i$ (e.g., a choice of medicine), and then it observes a real-valued reward $y_{t,i}$ (e.g., effectiveness of the prescribed medicine). In linear contextual bandits, the reward $y_{t,i}$ is a linear function of the unknown bandit parameter $\theta^* \in \mathbb{R}^d$ corrupted by $i.i.d$ mean-zero observation noise $\eta_{t,i}$, i.e., $y_{t,i} = \langle x_{t,i}, \theta^* \rangle + \eta_{t,i}$, where $x_{t,i} = \phi_i(c_{t,i}, a_{t,i})$ and $\phi_i : \mathcal{C}_i \times \mathcal{K}_i \to \mathbb{R}^d$ is a known function that maps a context-action pair to a $d$-dimensional real-valued feature vector. The goal of federated LCB is to minimize the cumulative *group* pseudo-regret defined

$$R_M(T) = \sum_{i=1}^{M} \sum_{t=1}^{T} \left[ \max_{a \in \mathcal{K}_i} \langle \phi_i(c_{t,i}, a), \theta^* \rangle - \langle x_{t,i}, \theta^* \rangle \right].$$

To achieve the goal, as in standard cross-silo FL, the agents are allowed to communicate with the central server following a star-shaped communication, i.e., each agent can communicate with the server by uploading and downloading data, but agents cannot communicate with each other directly. However, the communication process (i.e., both data and schedule) could also possibly incur privacy leakage for each user $t$ at each silo $i$, e.g., the sensitive context information $c_{t,i}$ and reward $y_{t,i}$.

To address this privacy risk, we resort to *differential privacy* (Dwork & Roth, 2014), a principled way to prove privacy guarantee against adversaries with arbitrary auxiliary information. In standard *cross-device* FL, the notion of privacy is often the client-level DP, which protects the identity of each participating client or device. However, it has limitations in cross-silo FL, where the protection

Table 1: Summary of main results. $\varepsilon > 0, \delta \in (0, 1)$ are privacy parameters.

| DP model | Privacy cost in regret | Communication | Reference |
|---|---|---|---|
| Central-JDP | $\widetilde{O}\left(\sqrt{MT}\frac{d^{3/4}\log^{1/4}(1/\delta)}{\sqrt{\varepsilon}}\right)$ | NA | Shariff & Sheffet (2018) |
| Silo-level LDP | $\widetilde{O}\left(\sqrt{T}\frac{(Md)^{3/4}\log^{1/4}(1/\delta)}{\sqrt{\varepsilon}}\right)$ | $O(\sqrt{MT})$ | Theorem 5.1 |
| SDP | $\widetilde{O}\left(\sqrt{MT}\frac{d^{3/4}\log^{1/4}(1/\delta)}{\sqrt{\varepsilon}}\right)$ | $O(\sqrt{MT})$ | Theorem 5.3 and 5.5 |

targets are users (e.g., patients) rather than participating silos or agents (e.g., hospitals). Also, in order to adopt client-level DP to cross-silo FL, one needs the server and other silos to be trustworthy, which is often not the case. Hence, recent studies (Lowy & Razaviyayn, 2021; Lowy et al., 2022; Liu et al., 2022; Dobbe et al., 2018) on cross-silo federated supervised learning have converged to a new privacy notion, *which requires that for each silo, all of its communication during the entire process is private ("indistinguishable") with respect to change of one local user of its own.* This allows one to protect *each user* within each silo without trustworthy server and other silos. In this paper, we adapt it to the setting of cross-silo federated contextual bandits and call it *silo-level LDP*[1].

Dubey & Pentland (2020) adopt a similar but somewhat weaker notion of privacy called *Federated DP* and takes the first step to tackle this important problem of private and federated linear contextual bandits (LCBs). In fact, the performance guarantees presented by the authors are currently the state-of-the-art for this problem. The proposed algorithm *claims* to protect the privacy of each user at each silo. Furthermore, given a privacy budget $\varepsilon > 0$, the claimed regret bound is $\widetilde{O}(\sqrt{MT/\varepsilon})$ with only $O(M \log T)$ communication rounds, which matches the regret of a super-single agent that plays for total $MT$ rounds. Unfortunately, in spite of being the state-of-the-art, the aforementioned privacy, regret and communication cost guarantees have fundamental gaps, as discussed below.

**Our contributions: identify privacy, regret, communication gaps in state-of-the-art (Dubey & Pentland, 2020).** In Section 3, we first show that the algorithm in (Dubey & Pentland, 2020) could leak privacy from the side channel of adaptive communication schedule, which depends on users' *non-private* local data. Next, we identify a mistake in total injected privacy noise in their regret analysis. Accounting for this miscalculation, the correct regret bound would amount to $\widetilde{O}(M^{3/4}\sqrt{T/\varepsilon})$, which is $M^{1/4}$ factor higher than the claimed one, and doesn't match regret performance of the super agent. Finally, we observe that due to the presence of privacy noise, its current analysis for $O(M \log T)$ communications no longer holds. To resolve these issues, we take the following two-step approach:

**(i) design a generic algorithmic and analytical framework.** In Section 4, we propose a generic federated LCB algorithm along with a flexible privacy protocol. Our algorithm adopts a fixed-batch schedule (rather than an adaptive one in Dubey & Pentland (2020)) that helps avoid privacy leakage from the side channel, as well as subtleties in communication analysis. Our privacy protocol builds on a distributed version of the celebrated tree-based algorithm (Chan et al., 2011; Dwork et al., 2010), enabling us to provide different privacy guarantees in a unified way. We further show that our algorithm enjoys a simple and generic analytical regret bound that only depends on the total amount of injected privacy noise under the required privacy constraints.

**(ii) prove regret guarantees under different privacy notions.** We build upon the above framework to study federated LCBs under two different privacy constraints. In Section 5.1, we consider silo-level LDP (a stronger notion of privacy than Federated DP of Dubey & Pentland (2020)) and establish privacy guarantee with a correct regret bound $\widetilde{O}(M^{3/4}\sqrt{T/\varepsilon})$ and communication cost $O(\sqrt{MT})$, hence fixing the gaps in Dubey & Pentland (2020). Next, to match the regret of a super single agent, we consider shuffle DP (SDP) (Cheu et al., 2019) in Section 5.2 and establish a regret bound of $\widetilde{O}(\sqrt{MT/\varepsilon})$. We provide two different techniques to achieve this – one that relies on a new result on privacy amplification via shuffling for DP mechanisms and the other that integrates a shuffle protocol for vector sums (Cheu et al., 2021) into the tree-based mechanism. See Table 1 for a summary.

**Related work.** In standard multi-armed bandits, where rewards are only sensitive data, different DP models including central (Mishra & Thakurta, 2015; Azize & Basu, 2022; Sajed & Sheffet, 2019),

---

[1]It appears under different names in prior work, e.g., silo-specific sample-level DP (Liu et al., 2022), inter-silo record-level DP (Lowy & Razaviyayn, 2021).

local (Ren et al., 2020) and distributed (Chowdhury & Zhou, 2022a; Tenenbaum et al., 2021), have been studied. In linear contextual bandits, where both contexts and rewards are sensitive, there is a line of work under central (Shariff & Sheffet, 2018), local (Zheng et al., 2020) and shuffle (Chowdhury & Zhou, 2022b; Garcelon et al., 2022; Tenenbaum et al., 2023) models of DP. Li et al. (2022); Hanna et al. (2022) study linear bandits without contexts protection. Dubey & Pentland (2020) is the first to consider federated LCBs under item-level privacy while Huang et al. (2023) study user-level privacy under some distributional assumptions; see Appendix A. Federated or distributed LCBs without privacy have also been studied (Wang et al., 2020; He et al., 2022a; Huang et al., 2021), where a common goal is to achieve the regret of a super single agent that plays $MT$ rounds while keeping communication cost minimal. Lowy & Razaviyayn (2021); Liu et al. (2022) study private cross-silo federated learning under supervised setting, whereas we focus on the sequential learning setting.

## 2 DIFFERENTIAL PRIVACY IN FEDERATED LCBS

We now formally introduce differential privacy in cross-silo federated contextual bandits. Let a dataset $D_i$ at each silo $i$ be given by a sequence of $T$ *unique* users $U_{1,i}, \ldots, U_{T,i}$. Each user $U_{t,i}$ is identified by her context information $c_{t,i}$ as well as reward responses she would give to all possible actions recommended to her. We say two datasets $D_i$ and $D'_i$ at silo $i$ are adjacent if they differ exactly in one participating user, i.e., $U_{\tau,i} \neq U'_{\tau,i}$ for some $\tau \in [T]$ and $U_{s,i} = U'_{s,i}$ for all $s \neq \tau$.

**Silo-level local differential privacy (LDP).** Consider a multi-round, cross-silo federated learning algorithm $\mathcal{Q}$. At each round $t$, each silo $i$ communicates a randomized message $Z_i^t$ of its data $D_i$ to the server, which may depend (due to collaboration) on previous randomized messages $Z_j^1, \ldots, Z_j^{t-1}$ from all other silos $j \neq i$. We allow $Z_i^t$ to be empty if there is no communication at round $t$. Let $Z_i = (Z_i^1, \ldots, Z_i^T)$ denote the full transcript of silo $i$'s communications with the server over $T$ rounds and $\mathcal{Q}_i$ the induced local mechanism in this process. Note that $Z_i$ is a realization of random messages generated according to the local mechanism $\mathcal{Q}_i$. We denote by $Z_{-i} = (Z_1, \ldots, Z_{i-1}, Z_{i+1}, \ldots, Z_M)$ the full transcripts of all but silo $i$. We assume that $Z_i$ is conditionally independent of $D_j$ for all $j \neq i$ given $D_i$ and $Z_{-i}$. With this notation, we have the following definition of silo-level LDP.

**Definition 2.1** (Silo-level LDP). A cross-silo federated learning algorithm $\mathcal{Q}$ with $M$ silos is said to be $(\varepsilon_i, \delta_i)_{i \in M}$ silo-level LDP if for each silo $i \in [M]$, it holds that

$$\mathbb{P}\big[\mathcal{Q}_i(Z_i \in \mathcal{E}_i | D_i, Z_{-i})\big] \leq e^{\varepsilon_i} \mathbb{P}\big[\mathcal{Q}_i(Z_i \in \mathcal{E}_i | D'_i, Z_{-i})\big] + \delta_i \,,$$

for all adjacent datasets $D_i$ and $D'_i$, and for all events $\mathcal{E}_i$ in the range of $\mathcal{Q}_i$. If $\varepsilon_i = \varepsilon$ and $\delta_i = \delta$ for all $i \in [M]$, we simply say $\mathcal{Q}$ is $(\varepsilon, \delta)$-silo-level LDP.

Roughly speaking, a silo-level LDP algorithm protects the privacy of each individual user (e.g., patient) within each silo in the sense that an adversary (which could either be the central server or other silos) cannot infer too much about any individual's sensitive information (e.g., context and reward) or determine whether an individual participated in the learning process.[2]

*Remark* 2.2 (Federated DP vs. Silo-level LDP). Dubey & Pentland (2020) consider a privacy notion called Federated DP (Fed-DP in short). As summarized in Dubey & Pentland (2020), Fed-DP requires "the action chosen by any agent must be sufficiently impervious (in probability) to any single pair $(x, y)$ from any other agent". Both silo-level LDP and Fed-DP are item-level DP as the neighboring relationship is defined by differing in one participating user. The key here is to note that silo-level DP implies Fed-DP by the post-processing property of DP, and thus it is a stronger notion of privacy. In fact, Dubey & Pentland (2020) claim to achieve Fed-DP by relying on privatizing the communicated data from each silo. However, as we shall see in Section 3, its proposed algorithm fails to privatize the adaptive synchronization schedule, which is the key reason behind privacy leakage in their algorithm.

**Shuffle differential privacy (SDP).** Another common DP notion for FL is SDP (Cheu et al., 2019), which has been widely studied in supervised learning (Lowy & Razaviyayn, 2021; Girgis et al., 2021; Lowy et al., 2022) to match the centralized utility performance. Motivated by this, we adapt it to FL-LCBs. Specifically, each silo $i \in [M]$ first applies a local randomizer $\mathcal{R}$ to its raw local data and sends the randomized output to a shuffler $\mathcal{S}$. The shuffler $\mathcal{S}$ permutes all the messages from all $M$ silos uniformly at random and sends those to the central server. Roughly speaking, SDP requires

---

[2]This is a notion of item-level DP. A comparison with standard local DP, central (joint)-DP and shuffle DP for single-agent LCBs is presented in Appendix G.2.

all the messages sent by the shuffler to be private ("indistinguishable") with respect to a single user change among all $MT$ users. This item-level DP is defined formally as follows.

**Definition 2.3** (SDP). Consider a cross-silo federated learning algorithm $\mathcal{Q}$ that induces a (randomized) mechanism $\mathcal{M}$ whose output is the collection of all messages sent by the shuffler during the entire learning process. Then, the algorithm $\mathcal{Q}$ is said to be $(\varepsilon, \delta)$-SDP if

$$\mathbb{P}\big[\mathcal{M}(D) \in \mathcal{E}\big] \leq e^{\varepsilon}\, \mathbb{P}\big[\mathcal{M}(D') \in \mathcal{E}\big] + \delta \,,$$

for all $\mathcal{E}$ in the range of $\mathcal{M}$ and for all adjacent datasets $D = (D_1, \ldots, D_M)$ and $D' = (D'_1, \ldots, D'_M)$ such that $\sum_{i=1}^{M} \sum_{t=1}^{T} \mathbb{1}_{\{U_{t,i} \neq U'_{t,i}\}} = 1$.

## 3   PRIVACY, REGRET AND COMMUNICATION GAPS IN STATE-OF-THE-ART

**Gap in privacy analysis.** We take a two-step approach to demonstrate the privacy issue in Dubey & Pentland (2020). To start with, we argue that Algorithm 1 in Dubey & Pentland (2020) fails to achieve silo-level LDP due to privacy leakage through the side channel of communication schedule (i.e., when agents communicate with the server). The key issue is that the adaptive communication schedule in the algorithm depends on users' *non-private* data. This fact can be utilized by an adversary or malicious silo $j$ to infer another silo $i$'s users' sensitive information, which violates the requirement of silo-level LDP. In that algorithm, *all* silos *synchronously* communicate with the server if

$$\exists \text{ some silo } i \in [M] : f(X_i, Z) > 0 \,, \tag{1}$$

where $f$ is some function, $X_i$ is non-private local data of silo $i$ since the last synchronization and $Z$ is all previously synchronized data. Crucially, the form of $f$ and the rule (1) are public information, known to all silos even before the algorithm starts. This local and non-private data-dependent communication rule in (1) causes privacy leakage, as illustrated below with a toy example.

**Example 3.1** (Privacy leakage). Consider two silos $i$ and $j$ following Algorithm 1 in Dubey & Pentland (2020). After the first round, $X_i$ is the data of the first user in silo $i$ (say Alice), $X_j$ is the data of the first user in silo $j$ (say Bob) and $Z$ is zero. Let communication is triggered at the end of first round and assume $f(X_j, 0) \leq 0$. Since the rule (1) is public, silo $j$ can infer that $f(X_i, 0) > 0$, i.e. the communication is triggered by silo $i$. Since $f$ is also public knowledge, silo $j$ can utilize this to infer some property of $X_i$. Hence, by observing *only* the communication signal (even without looking at the data), silo $j$ can infer sensitive data of Alice. In fact, the specific form of $f$ in Dubey & Pentland (2020) allows silo $j$ to infer context information of Alice (details in Appendix B).

This example shows that Algorithm 1 in Dubey & Pentland (2020) does not satisfy silo-level LDP, implying their proof for Fed-DP guarantee via post-processing of silo-level LDP does not hold. However, it does not imply that this algorithm fails to satisfy Fed-DP, which is a weaker notion than silo-level LDP. Nevertheless, by leveraging Example 3.1, one can show that this algorithm indeed fails to guarantee Fed-DP. To see this, recall the definition of Fed-DP from Remark 2.2. In the context of Example 3.1, it translates to silo $j$ selecting similar actions for its users when a single user in silo $i$ changes. Specifically, if the first user in silo $i$ changes from Alice to say, Tracy, Fed-DP mandates that all $T$ actions suggested by silo $j$ to its local $T$ users remain "indistinguishable". This, in turn, implies that the communicated data from silo $i$ must remain "indistinguishable" at silo $j$ for each $t \in [T]$. This is because the actions at silo $j$ are chosen *deterministically* based on its local data as well as on the communicated data from silo $i$, and the local data at silo $j$ remains unchanged. However, in Algorithm 1 of Dubey & Pentland (2020), the communicated data from silo $i$ is not guaranteed to remain "indistinguishable" as synchronization depends on non-private local data ($X_i$ in (1)). In other words, without additional privacy noise added to $X_i$ in (1), the change from Alice to Tracy could affect the *existence of synchronization* at round $t \geq 1$. Consequently, under these two neighboring situations (e.g. Alice vs. Tracy), the communicated data from silo $i$ could differ significantly at round $t + 1$. As a result, the action chosen at round $t + 1$ in silo $j$ can be totally different violating Fed-DP. This holds true even if silo $i$ injects noise while communicating its data (as done in Dubey & Pentland (2020)) due to a large change of non-private communicated data (see Appendix B for details).

**Gaps in regret and communication analysis.** We now turn to regret and communication analysis of Dubey & Pentland (2020), which has fundamental gaps that lead to incorrect conclusions in the end. First, the reported privacy cost in regret bound is $\tilde{O}(\sqrt{MT/\varepsilon})$ (ignoring dependence on dimension $d$), which leads to the conclusion that federated LCBs across $M$ silos under silo-level LDP can achieve the same order of regret as in the centralized setting (i.e., when a super single agent

---

**Algorithm 1** Private-FedLinUCB

---

1: **Parameters:** Batch size $B \in \mathbb{N}$, regularization $\lambda > 0$, confidence radii $\{\beta_{t,i}\}_{t \in [T], i \in [M]}$, feature map $\phi_i : \mathcal{C}_i \times \mathcal{K}_i \to \mathbb{R}^d$, privacy protocol $\mathcal{P} = (\mathcal{R}, \mathcal{S}, \mathcal{A})$

2: **Initialize:** $W_i = 0, U_i = 0$ for all agents $i \in [M]$, $\widetilde{W}_{\text{syn}} = 0, \widetilde{U}_{\text{syn}} = 0$

3: **for** $t = 1, \ldots, T$ **do**

4:     **for** each agent $i = 1, \ldots, M$ **do**

5:         Receive context $c_{t,i}$; compute $V_{t,i} = \lambda I + \widetilde{W}_{\text{syn}} + W_i$ and $\widehat{\theta}_{t,i} = V_{t,i}^{-1}(\widetilde{U}_{\text{syn}} + U_i)$

6:         Play action $a_{t,i} = \text{argmax}_{a \in \mathcal{K}_i} \langle \phi_i(c_{t,i}, a), \widehat{\theta}_{t,i} \rangle + \beta_{t,i} \|\phi_i(c_{t,i}, a)\|_{V_{t,i}^{-1}}$; observe reward $y_{t,i}$

7:         Set $x_{t,i} = \phi_i(c_{t,i}, a_{t,i})$, $U_i = U_i + x_{t,i} y_{t,i}$ and $W_i = W_i + x_{t,i} x_{t,i}^\top$

8:     **end for**

9:     **if** $t \bmod B = 0$ **then**

10:         // Local randomizer $\mathcal{R}$ at *all* agents $i \in [M]$

11:         Send randomized messages $R_{t,i}^{\text{bias}} = \mathcal{R}^{\text{bias}}(U_i)$ and $R_{t,i}^{\text{cov}} = \mathcal{R}^{\text{cov}}(W_i)$ to $\mathcal{S}$

12:         // Third party $\mathcal{S}$

13:         Shuffle (or, not) all messages $S_t^{\text{bias}} = \mathcal{S}(\{R_{t,i}^{\text{bias}}\}_{i \in [M]})$ and $S_t^{\text{cov}} = \mathcal{S}(\{R_{t,i}^{\text{cov}}\}_{i \in [M]})$

14:         // Analyzer $\mathcal{A}$ at the server

15:         Compute private synchronized statistics $\widetilde{U}_{\text{syn}} = \mathcal{A}^{\text{bias}}(S_t^{\text{bias}})$ and $\widetilde{W}_{\text{syn}} = \mathcal{A}^{\text{cov}}(S_t^{\text{cov}})$

16:         // All agents $i \in [M]$

17:         Receive $\widetilde{W}_{\text{syn}}$ and $\widetilde{U}_{\text{syn}}$ from the server and reset $W_i = 0, U_i = 0$

18:     **end if**

19: **end for**

---

plays $MT$ rounds). However, in the proposed analysis, the total amount of injected privacy noise is miscalculated. In particular, variance of total noise needs to be $M\sigma^2$ rather than the proposed value of $\sigma^2$. This is due to the fact that each silo injects Gaussian noise with variance $\sigma^2$ when sending out local data which amounts to total $M\sigma^2$ noise at the server. Accounting for this correction, the cost of privacy becomes $\tilde{O}(M^{3/4}\sqrt{T/\varepsilon})$, which is $O(M^{1/4})$ factor worse than the claimed one. Hence, we conclude that Algorithm 1 in Dubey & Pentland (2020) cannot achieve the same order of regret as in centralized setting. Second, the proposed analysis to show $O(\log T)$ communication rounds for the *data-adaptive* schedule (1) under privacy constraint essentially follows from the non-private one of Wang et al. (2020). Unfortunately, due to privacy noise, this direct approach no longer holds, and hence the reported logarithmic cost stands ungrounded (details in Appendix B).

## 4 OUR APPROACH

To address the issues in Dubey & Pentland (2020), we introduce a generic algorithm for private, federated linear contextual bandits (Algorithm 1) and a flexible privacy protocol (Algorithm 2). This helps us (a) derive correct privacy, regret, and communication results under silo-level LDP (and hence under Fed-DP) (Section 5.1), and (b) achieve the same order of regret as in centralized setting under SDP (Section 5.2). Throughout the paper, we make the following standard assumptions in LCBs.

**Assumption 4.1** (Boundedness (Shariff & Sheffet, 2018)). The rewards are bounded, i.e., $y_{t,i} \in [0, 1]$ for all $t \in [T]$ and $i \in [M]$. Moreover, $\|\theta^*\|_2 \le 1$ and $\sup_{c,a} \|\phi_i(c, a)\|_2 \le 1$ for all $i \in [M]$.

### 4.1 ALGORITHM: PRIVATE FEDERATED LINUCB

We build upon the celebrated LinUCB algorithm (Abbasi-Yadkori et al., 2011) by adopting a *fixed-batch* schedule for synchronization among agents and designing a privacy protocol $\mathcal{P}$ (Algorithm 2) for both silo-level LDP and SDP . At each round $t$, each agent $i$ recommends an action $a_{t,i}$ to each local user following *optimism in the face of uncertainty* principle. First, the agent computes a local estimate $\widehat{\theta}_{t,i}$ based on *all* available data to her, which includes previously synchronized data from all agents as well as her own new local data (line 5 of Algorithm 1). Then, the action $a_{t,i}$ is selected based on the LinUCB decision rule (line 6), where a proper radius $\beta_{t,i}$ is chosen to balance between exploration and exploitation. After observing the reward $y_{t,i}$, each agent accumulates her own local data (bias vector $x_{t,i} y_{t,i}$ and covariance matrix $x_{t,i} x_{t,i}^\top$) and stores them in $U_i$ and $W_i$, respectively

**Algorithm 2** $\mathcal{P}$, a privacy protocol used in Algorithm 1

1: **Procedure:** Local Randomizer $\mathcal{R}$ at each agent
2: //Input: stream data $(\gamma_1, \ldots, \gamma_K)$, $\varepsilon > 0, \delta \in (0, 1]$
3:     **for** $k = 1, \ldots, K$ **do**
4:         Express $k$ in binary form: $k = \sum_j \mathrm{Bin}_j(k) \cdot 2^j$
5:         Find index of first one $i_k = \min\{j : \mathrm{Bin}_j(k) = 1\}$
6:         Compute p-sum $\alpha_{i_k} = \sum_{j < i_k} \alpha_j + \gamma_k$
7:         Output $\widehat{\alpha}_k = \alpha_{i_k} + \mathcal{N}(0, \sigma_0^2 I)$
8:     **end for**
9: **Procedure:** Analyzer $\mathcal{A}$ at server
10: //Input : data from $\mathcal{S} : (\widehat{\alpha}_{k,1}, \ldots, \widehat{\alpha}_{k,M}), k \in [K]$
11:     **for** $k = 1, \ldots, K$ **do**
12:         Express $k$ in binary and find index of first one $i_k$
13:         Add noisy p-sums of all agents: $\widetilde{\alpha}_{i_k} = \sum_{i=1}^M \widehat{\alpha}_{k,i}$
14:         Output: $\widetilde{s}_k = \sum_{j : \mathrm{Bin}_j(k) = 1} \widetilde{\alpha}_j$
15:     **end for**

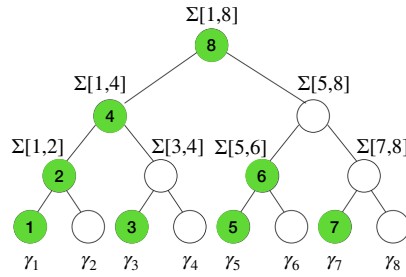

Figure 1: Illustration of the tree-based algorithm. Each leaf node is the stream data and each internal node is a p-sum $\Sigma[i, j] = \sum_{l=i}^j \gamma_l$. The green node corresponds to the newly computed p-sum at each $k$, i.e., $\alpha_{i_k}$ in Algorithm 2.

(line 7). A communication is triggered between agents and central server whenever a batch ends – we assume w.l.o.g. total rounds $T$ is divisible by batch size $B$ (line 9). During this process, a protocol $\mathcal{P} = (\mathcal{R}, \mathcal{S}, \mathcal{A})$ assists in aggregating local data among all agents while guaranteeing privacy properties (to be discussed in detail soon). After communication, each agent receives latest synchronized data $\widetilde{W}_{\mathrm{syn}}, \widetilde{U}_{\mathrm{syn}}$ from the server (line 17). Here, for any $t = kB, k \in [T/B]$, $\widetilde{W}_{\mathrm{syn}}$ represents noisy version of all covariance matrices up to round $t$ from all agents (i.e., $\sum_{i=1}^M \sum_{s=1}^t x_{s,i} x_{s,i}^\top$) and similarly, $\widetilde{U}_{\mathrm{syn}}$ represents noisy version of all bias vectors $\sum_{i=1}^M \sum_{s=1}^t x_{s,i} y_{s,i}$. Finally, each agent resets $W_i$ and $U_i$ so that they can be used to accumulate new local data for the next batch. Note that Algorithm 1 uses a fixed-batch (data-independent) communication schedule rather than the adaptive, data-dependent one in Dubey & Pentland (2020), which allows us to resolve privacy and communication issues.

## 4.2 PRIVACY PROTOCOL

We now turn to our privacy protocol $\mathcal{P}$ (Algorithm 2), which helps to aggregate data among all agents under privacy constraints. The key component of $\mathcal{P}$ is a *distributed* version of the classic tree-based algorithm, which was originally designed for continual release of private sum statistics (Chan et al., 2011; Dwork et al., 2010). That is, given a stream of (multivariate) data $\gamma = (\gamma_1, \ldots, \gamma_K)$, one aims to release $s_k = \sum_{l=1}^k \gamma_l$ *privately* for *all* $k \in [K]$. The tree-based mechanism constructs a complete binary tree $\mathcal{T}$ in online manner. The leaf nodes contain data $\gamma_1$ to $\gamma_K$, and internal nodes contain the sum of all leaf nodes in its sub-tree, see Fig. 1 for an illustration. For any new arrival data $\gamma_k$, it only releases a tree node privately, which corresponds to a noisy partial sum (p-sum) between two time indices. As an example, take $k = 6$, and hence the new arrival is $\gamma_6$. The tree-based mechanism first computes the p-sum $\sum[5, 6] = \gamma_5 + \gamma_6$ (line 6 in Algorithm 2). Then, it adds a Gaussian noise with appropriate variance $\sigma_0^2$ to $\sum[5, 6]$ and releases the noisy p-sum (line 7). Finally, to compute the prefix sum statistic $\sum[1, 6]$ privately, it simply adds noisy p-sums for $\sum[1, 4]$ and $\sum[5, 6]$, respectively. Reasons behind releasing and aggregating p-sums are that (i) each data point $\gamma_k$ only affects at most $1 + \log K$ p-sums (useful for privacy) and (ii) each sum statistic $\sum[1, k]$ only involves at most $1 + \log k$ p-sums (useful for utility).

Our privacy protocol $\mathcal{P} = (\mathcal{R}, \mathcal{S}, \mathcal{A})$ breaks down the above classic mechanism of releasing and aggregating p-sums into a local randomizer $\mathcal{R}$ at each agent and an analyzer $\mathcal{A}$ at the server, separately, while allowing for a possible shuffler in between to amplify privacy. For each $k$, the local randomizer $\mathcal{R}$ at each agent computes and releases the noisy p-sum to a third-party $\mathcal{S}$ (lines 4-7). $\mathcal{S}$ can either be a shuffler that permutes the data uniformly at random (for SDP) or can simply be an identity mapping (for silo-level LDP). It receives a total of $M$ noisy p-sums, one from each agent, and sends them to the central server. The analyzer $\mathcal{A}$ at the server first adds these $M$ new noisy p-sums to synchronize them (line 13). It then privately releases the synchronized prefix sum by adding up all relevant synchronized p-sums as discussed in above paragraph (line 14). Finally, we employ $\mathcal{P}$ to Algorithm 1 by observing that local data $\gamma_{k,i}$ for batch $k$ and agent $i$ consists of bias vectors $\gamma_{k,i}^{\mathrm{bias}} = \sum_{t=(k-1)B+1}^{kB} x_{t,i} y_{t,i}$ and covariance matrices $\gamma_{k,i}^{\mathrm{cov}} = \sum_{t=(k-1)B+1}^{kB} x_{t,i} x_{t,i}^\top$, which are stored

in $U_i$ and $W_i$ respectively. We denote the randomizer and analyzer for bias vectors as $\mathcal{R}^{\text{bias}}$ and $\mathcal{A}^{\text{bias}}$, and for covariance matrices as $\mathcal{R}^{\text{cov}}$ and $\mathcal{A}^{\text{cov}}$ in Algorithm 1.

## 5 THEORETICAL RESULTS

### 5.1 FEDERATED LCBS UNDER SILO-LEVEL LDP

We first present the performance of Algorithm 1 under silo-level LDP, hence fixing the privacy, regret and communication issues of the state-of-the-art algorithm in Dubey & Pentland (2020). The key idea is to inject Gaussian noise with proper variance ($\sigma_0^2$ in Algorithm 2) when releasing a p-sum such that all the released p-sums up to any batch $k \in [K]$ is $(\varepsilon, \delta)$-DP for any agent $i \in [M]$. Then, by Definition 2.1, it achieves silo-level LDP. Note that in this case, there is no shuffler, which is equivalent to the fact that the third party $\mathcal{S}$ in $\mathcal{P}$ is simply an identity mapping, denoted by $\mathcal{I}$. The following result states this formally, with proof deferred to Appendix E.

**Theorem 5.1** (Performance under silo-level LDP). *Fix batch size $B$, privacy budgets $\varepsilon > 0$, $\delta \in (0, 1)$. Let $\mathcal{P} = (\mathcal{R}, \mathcal{I}, \mathcal{A})$ be a protocol given by Algorithm 2 with parameters $\sigma_0^2 = 8\kappa \cdot \frac{(\log(2/\delta) + \varepsilon)}{\varepsilon^2}$, where $\kappa = 1 + \log(T/B)$. Then, under Assumption 4.1, Algorithm 1 instantiated with $\mathcal{P}$ satisfies $(\varepsilon, \delta)$-silo-level LDP. Moreover, for any $\alpha \in (0, 1]$, there exist choices of $\lambda$ and $\{\beta_{t,i}\}_{t,i}$ such that, with probability at least $1 - \alpha$, it enjoys a group regret*

$$R_M(T) = O\left(dMB \log T + d\sqrt{MT} \log(MT/\alpha)\right) + \widetilde{O}\left(\sqrt{T} \frac{(Md)^{3/4} \log^{1/4}(1/\delta)}{\sqrt{\varepsilon}} \log^{1/4}(T/B\alpha)\right).$$

The first term in the above regret bound doesn't depend on privacy budgets $\varepsilon, \delta$, and serves as a representative regret bound for federated LCBs without privacy constraint. The second term is the dominant one which depends on $\varepsilon, \delta$ and denotes the cost of privacy due to injected noise.

**Corollary 5.2.** *Setting $B = \sqrt{T/M}$, Algorithm 1 achieves $\widetilde{O}\left(d\sqrt{MT} + \sqrt{T} \frac{(Md)^{3/4} \log^{1/4}(1/\delta)}{\sqrt{\varepsilon}}\right)$ group regret, with total $\sqrt{MT}$ synchronizations under $(\varepsilon, \delta)$-silo-level LDP.*

**Comparison with Dubey & Pentland (2020).** First, we avoid the privacy leakage by adopting data-independent synchronization. However, this leads to an $O(\sqrt{T})$ communication cost. It remains open to design a (correct) data-adaptive schedule with logarithmic cost; details in Appendix D. We also show that privacy cost scales as $O(M^{3/4})$ with number of agents $M$, correcting the reported $\sqrt{M}$ scaling. Next, we compare our result with that of a super single agent running for $MT$ rounds under the central model of DP (i.e., where the central server is trusted), which serves as a benchmark for our results. As shown in Shariff & Sheffet (2018), the total regret for such a single agent is $\widetilde{O}\left(d\sqrt{MT} + \sqrt{MT} \frac{d^{3/4} \log^{1/4}(1/\delta)}{\sqrt{\varepsilon}}\right)$. Comparing this with Corollary 5.2, we observe that the privacy cost of federated LCBs under silo-level LDP is a multiplicative $M^{1/4}$ factor higher than a super agent under the central model. This motivates us to consider SDP in next section.

### 5.2 FEDERATED LCBS UNDER SDP

We now aim to close the above $M^{1/4}$ gap in the privacy cost under silo-level LDP compared to that achieved by a super single agent (with a truseted central server). To do so, we consider federated LCBs under SDP, which still enjoys the nice feature of silo-level LDP that the central server is not trusted. Thanks to our flexible protocol $\mathcal{P}$, the only change needed compared to silo-level LDP is the introduction of a shuffler $\mathcal{S}$ to amplify privacy and adjustment of the privacy noise $\sigma_0^2$ accordingly.

**Theorem 5.3** (Performance under SDP via amplification). *Fix batch size $B$ and let $\kappa = 1 + \log(T/B)$. Let $\mathcal{P} = (\mathcal{R}, \mathcal{S}, \mathcal{A})$ be a protocol given by Algorithm 2. Then, under Assumption 4.1, there exist constants $C_1, C_2 > 0$ such that for any $\varepsilon \leq \frac{\sqrt{\kappa}}{C_1 T \sqrt{M}}$, $\delta \leq \frac{\kappa}{C_2 T}$, Algorithm 1 instantiated with $\mathcal{P}$ and $\sigma_0^2 = O\left(\frac{2\kappa \log(1/\delta) \log(\kappa/(\delta T)) \log(M\kappa/\delta)}{\varepsilon^2 M}\right)$, satisfies $(\varepsilon, \delta)$-SDP. Moreover, for any $\alpha \in (0, 1]$, there exist choices of $\lambda$ and $\{\beta_{t,i}\}_{t,i}$ such that, with a probability at least $1 - \alpha$, it enjoys a group regret*

$$R_M(T) = O\left(dMB \log T + d\sqrt{MT} \log(MT/\alpha)\right) + \widetilde{O}\left(d^{3/4} \sqrt{MT} \frac{\log^{3/4}(M\kappa/\delta)}{\sqrt{\varepsilon}} \log^{1/4}(T/B\alpha)\right).$$

**Corollary 5.4.** *Setting $B = \sqrt{T/M}$, Algorithm 1 achieves $\widetilde{O}\left(d\sqrt{MT} + d^{3/4}\sqrt{MT} \frac{\log^{3/4}(M\kappa/\delta)}{\sqrt{\varepsilon}}\right)$ group regret, with total $\sqrt{MT}$ synchronizations under $(\varepsilon, \delta)$-SDP.*

Corollary 5.4 asserts that privacy cost of federated LCBs under SDP matches that of a super single agent under central DP (up to a log factor in $T, M, \delta$).

**Comparison with existing SDP analysis.** Note that the above result doesn't directly follow from prior amplification results (Feldman et al., 2022; Erlingsson et al., 2019; Cheu et al., 2019; Balle et al., 2019), which show that shuffling outputs of $M$ $(\varepsilon, \delta)$-LDP algorithms achieve roughly $1/\sqrt{M}$ factor amplification in privacy for small $\varepsilon$ – the key to close the aforementioned gap in privacy cost. However, these amplification results apply *only* when each mechanism is LDP *in the standard sense, i.e., they operate on a dataset of size $n = 1$*. This doesn't hold in our case since the dataset at each silo is a stream of $T$ points. Lowy & Razaviyayn (2021) adopt group privacy to handle the case of $n > 1$, which can amplify any general DP mechanism but comes at the expense of a large increase in $\delta$. To avoid this, we prove a *new amplification lemma* specific to Gaussian DP mechanisms operating on datasets with size $n > 1$. This helps us achieve the required $1/\sqrt{M}$ amplification in $\varepsilon$ while keeping the increase in $\delta$ in check. The key idea behind our new lemma is to directly analyze the sensitivity when creating "clones" as in Feldman et al. (2022), but now by accounting for the fact that all $n > 1$ points can be different (see Appendix F for formal statement and proof of the lemma).

### 5.2.1 SDP GUARANTEE FOR A WIDE RANGE OF PRIVACY PARAMETERS

One limitation of attaining SDP via amplification is that the privacy guarantee holds only for small values of $\varepsilon, \delta$ (see Theorem 5.3). In this subsection, we propose an alternative privacy protocol to resolute this limitation. This new protocol leverages the same binary tree structure as in Algorithm 2 for releasing and aggregating p-sums, but it employs different local randomizers and analyzers for computing (noisy) synchronized p-sums of bias vectors and covariance matrices ($\widetilde{\alpha}_{i_k}$ in Algorithm 2). Specifically, it applies the vector sum mechanism $\mathcal{P}_{\text{Vec}}$ of Cheu et al. (2021), which essentially take $n$ vectors as inputs and outputs their noisy sum. Here privacy is ensured by injecting suitable binomial noise to a fixed-point encoding of each vector entry, which depends on $\varepsilon, \delta$ and $n$.

In our case, one cannot directly aggregate $M$ p-sums using $\mathcal{P}_{\text{Vec}}$ with $n = M$. This is because each p-sum would then have a large norm ($O(T)$ at the worst case), which would introduce a large amount of privacy noise (cf. Theorem 3.2 in Cheu et al. (2021)), resulting in worse utility (regret). Instead, we first expand each p-sum resulting in a set of points such that each with $O(1)$ norm. Then, we aggregate all of those data points using $\mathcal{P}_{\text{Vec}}$ mechanism (one each for bias vectors and covariance matrices). For example, consider summing bias vectors during batch $k = 6$ and refer back to Fig. 1 for illustration. Here, the p-sum for each agent is given by $\sum[5, 6] = \gamma_5 + \gamma_6$ (see line 6 in Algorithm 2), the expansion of which results in $2B$ bias vectors ($B$ each for batch 5 and 6). A noisy sum of $n = 2BM$ bias vectors is then computed using $\mathcal{P}_{\text{Vec}}$. We denote the entire mechanism as $\mathcal{P}_{\text{Vec}}^{\mathcal{T}}$ – see Algorithm 5 in Appendix F.2 for pseudo-code and a complete description.

Now, the key intuition behind using $\mathcal{P}_{\text{Vec}}$ as a building block is that it allows us to compute private vector sums under the shuffle model using nearly the same amount of noise as in the central model. In other words, it "simulates" the privacy noise introduced in vector summation under the central model using a shuffler. This, in turn, helps us match the regret of the centralized setting while guaranteeing (strictly stronger) SDP. Specifically, we have the same order of regret as in Theorem 5.3, but now it holds for a wide range of privacy budgets $\varepsilon, \delta$ as presented below. Proof is deferred to Appendix F.

**Theorem 5.5** (Performance under SDP via vector sum). *Fix batch size $B$ and let $\kappa = 1 + \log(T/B)$. Let $\mathcal{P}_{\text{Vec}}^{\mathcal{T}}$ be a privacy protocol given by Algorithm 5. Then, under Assumption 4.1, there exist parameter choices of $\mathcal{P}_{\text{Vec}}^{\mathcal{T}}$ such that for any $\varepsilon \leq 60\sqrt{2\kappa \log(2/\delta)}$ and $\delta \leq 1$, Algorithm 1 instantiated with $\mathcal{P}_{\text{Vec}}^{\mathcal{T}}$ satisfies $(\varepsilon, \delta)$-SDP. Moreover, for any $\alpha \in (0, 1]$, there exist choices of $\lambda$ and $\{\beta_{t,i}\}_{t,i}$ such that, with a probability at least $1 - \alpha$, it enjoys a group regret*

$$R_M(T) = O\left(dMB \log T + d\sqrt{MT} \log(MT/\alpha)\right) + \widetilde{O}\left(d^{3/4}\sqrt{MT} \frac{\log^{3/4}(\kappa d^2/\delta)}{\sqrt{\varepsilon}} \log^{1/4}(T/B\alpha)\right).$$

*Remark* 5.6 (Importance of communicating P-sums). A key technique behind closing the regret gap under SDP is to communicate and shuffle *only* the p-sums rather than prefix sums. With this we can ensure that each data point (bias vector/covariance matrix) participates only in $O(\log K)$ shuffle mechanisms (rather than in $O(K)$ mechanisms if we communicate and shuffle prefix-sums). This helps us keep the final privacy cost in check after adaptive composition. In other words, one cannot simply use shuffling to amplify privacy of Algorithm 1 in Dubey & Pentland (2020) to close the regret gap (even ignoring its privacy and communication issues), since it communicates prefix sums at each

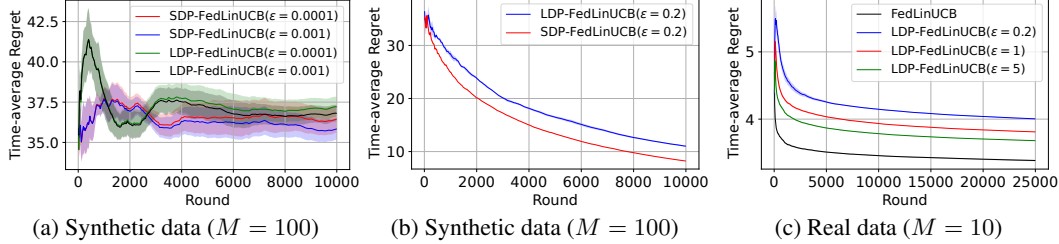

(a) Synthetic data ($M = 100$)    (b) Synthetic data ($M = 100$)    (c) Real data ($M = 10$)

Figure 2: Comparison of time-average group regret for LDP-FedLinUCB (silo-level LDP), SDP-FedLinUCB (shuffle model) and FedLinUCB (non-private) under varying privacy budgets $\varepsilon, \delta$ on (a, b) synthetic Gaussian bandit instance and (c) bandit instance generated from MSLR-WEB10K Learning to Rank dataset.

synchronization. This again highlights the algorithmic novelty of our privacy protocols (Algorithms 2 and 5), which could be of independent interest. See Appendix F for further details.

## 6   SIMULATION RESULTS AND CONCLUSIONS

We evaluate regret performance of Algorithm 1 under silo-level LDP and SDP, which we abbreviate as LDP-FedLinUCB and SDP-FedLinUCB, respectively. We fix confidence level $\alpha = 0.01$, batchsize $B = 25$ and study comparative performances under varying privacy budgets $\varepsilon, \delta$. We plot time-averaged group regret $\mathrm{Reg}_M(T)/T$ in Figure 2 by averaging results over 25 parallel runs.

**Synthetic bandit instance.** We simulate a LCB instance with a parameter $\theta^*$ of dimension $d = 10$ and $|\mathcal{K}_i| = 100$ actions for each of the $M$ agents. Similar to Vaswani et al. (2020), we generate $\theta^*$ and feature vectors by sampling a $(d-1)$-dimensional vectors of norm $1/\sqrt{2}$ uniformly at random, and append it with a $1/\sqrt{2}$ entry. Rewards are corrupted with Gaussian $\mathcal{N}(0, 0.25)$ noise.

**Real-data bandit instance.** We generate bandit instances from Microsoft Learning to Rank dataset (Qin & Liu, 2013). Queries form contexts $c$ and actions $a$ are the available documents. The dataset contains 10K queries, each with up to 908 judged documents on scale of $\mathrm{rel}(c, a) \in \{0, 1, 2\}$. Each pair $(c, a)$ has a feature vector $\phi(c, a)$, which is partitioned into title and body features of dimensions 57 and 78, respectively. We first train a lasso regression model on title features to predict relevances from $\phi$, and take this model as the parameter $\theta^*$ with $d = 57$. Next, we divide the queries equally into $M = 10$ agents and assign corresponding feature vectors to the agents. This way, we obtain a federated LCB instance with 10 agents, each with number of actions $|\mathcal{K}_i| \leq 908$.

**Observations.** In Fig. 2(a), we compare performance of LDP-FedLinUCB and SDP-FedLinUCB (with amplification based privacy protocol $\mathcal{P}$) on synthetic bandit instance with $M = 100$ agents under privacy budget $\delta = 0.0001$ and $\varepsilon = 0.001$ or $0.0001$. We observe that regret of SDP-FedLinUCB is less than LDP-FedLinUCB for both values of $\varepsilon$, which is consistent with theoretical results. Here, we only work with small privacy budgets since the privacy guarantee of Theorem 5.3 holds for $\varepsilon, \delta \ll 1$. Instead, in Fig. 2(b), we consider higher privacy budgets as suggested in Theorem 5.5 (e.g. $\varepsilon = 0.2$, $\delta = 0.1$) and compare the regret performance of LDP-FedLinUCB and SDP-FedLinUCB (with vecor-sum based privacy protocol $\mathcal{P}_{\mathrm{vec}}^{\mathcal{T}}$). As expected, we observe that regret of SDP-FedLinUCB decreases faster than that of LDP-FedLinUCB. Next, we benchmark the performance of Algorithm 1 under silo-level LDP (i.e. LDP-FedLinUCB) against a non-private Federated LCB algorithm with fixed communication schedule, building on Abbasi-Yadkori et al. (2011) and refer as FedLinUCB. In Fig. 2(c), we demonstrate the cost of privacy under silo-level LDP on real-data bandit instance by varying $\varepsilon$ in the set $\{0.2, 1, 5\}$ while keeping $\delta$ fixed to 0.1. We observe that regret of LDP-FedLinUCB decreases and comes closer to that of FedLinUCB as $\varepsilon$ increases (i.e., level of privacy protection decreases). A similar regret behavior is noticed under SDP also (postponed to Appendix H).

**Concluding Remarks.** We conclude with some discussions. First, our adversary model behind silo-level LDP excludes malicious users within the same silo. If one is also interested in protecting against adversary users within the same silo, a simple tweak of Algorithm 1 would suffice (see Appendix G.3). With this, one can not only protect against colluding silos (as in silo-level LDP), but also against colluding users within the same silo (as in central JDP). Next, we assume that all $MT$ users are unique in our algorithms. In practice, a user can participate in multiple rounds within the same silo or across different silos; see Appendix G.4. Finally, for future work, a challenging task is to achieve $O(\log T)$ communication cost with correct privacy and regret guarantees; see Appendix D for further discussions.

## ACKNOWLEDGEMENTS

XZ is supported in part by NSF CNS-2153220 and CNS-2312835. XZ would like to thank Abhimanyu Dubey for discussions on the work of Dubey & Pentland (2020). XZ would also like to thank Andrew Lowy and Ziyu Liu for insightful discussions on the privacy notion for cross-silo federated learning. XZ would thank Vitaly Feldman and Audra McMillan for the discussion on some subtleties behind "hiding among the clones" in Feldman et al. (2022).

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

## A    MORE DETAILS ON RELATED WORK

Private bandit learning has also been studied beyond linear settings, such as kernel bandits (Zhou & Tan, 2021; Dubey, 2021; Li et al., 2023). It is worth noting that Chowdhury & Zhou (2022a) also presents optimal private regret bounds under all three DP models (i.e., central, local, and distributed) in bandits while only relying on discrete privacy noise, hence avoiding the privacy leakage of continuous privacy noise on finite computers due to floating point arithmetic.

Recently, Huang et al. (2023) took the pioneering step to study user-level privacy for federated LCBs, establishing both regret upper bounds and lower bounds. In contrast to our item-level DP (e.g., silo-level LDP), user-level DP in Huang et al. (2023) roughly requires that even replacing the whole local history at any agent, the central server's broadcast message should be close across the whole learning period. This notion is more likely to be preferred in cross-device FL settings where the protection target is the device (agent). In addition to this, there are several other key differences compared to our work. First, they deal with linear bandits with stochastic contexts under additional distribution coverage assumptions (rather than the arbitrary adversary contexts in our case). In fact, it has been shown in Huang et al. (2021) that some assumption on the context distribution is necessary for a sublinear regret under user-level DP. Second, due to this stochastic context and some coverage conditions on contexts, an exponentially growing batch schedule can be applied in their case. In contrast, under the adversary context case, it is unclear to us how to apply the same technique to derive a sublinear regret.

## B    MORE DISCUSSIONS ON GAPS IN SOTA

In this section, we provide more details on the current gaps in Dubey & Pentland (2020), especially on privacy violation and communication cost. It turns out that both gaps come from the fact that an adaptive communication schedule is employed in Dubey & Pentland (2020).

### B.1    MORE ON VIOLATION OF SILO-LEVEL LDP

As shown in the main paper, Algorithm 1 in Dubey & Pentland (2020) does not satisfy silo-level LDP. To give a more concrete illustration of privacy leakage, we now specify the form of $f^3$, local data $X_i$ and synchronized data $Z$ in (1) according to Dubey & Pentland (2020). In particular, a communication is triggered at round $t$ if for any silo $i$, it holds that

$$(t-t') \log \left[ \frac{\det \left( Z + \sum_{s=t'+1}^{t} x_{s,i} x_{s,i}^\top + \lambda_{\min} I \right)}{\det \left( Z + \lambda_{\min} I \right)} \right] > D, \tag{2}$$

where $t'$ is the latest synchronization time before $t$, $Z$ is all synchronized (private) covariance matrices up to time $t'$, $\lambda_{\min} > 0$ is some regularization constant (which depends on privacy budgets $\varepsilon, \delta$) and $D > 0$ is some suitable threshold (which depends on number of silos $M$).

With the above explicit form in hand, we can give a more concrete discussion of Example 3.1. A communication is triggered at round $t = 1$ if $\det \left( x_{1,m} x_{1,m}^\top + \lambda_{\min} I \right) > \det \left( \lambda_{\min} I \right) e^D$ holds for any silo $m$. This implies that $(\lambda_{\min} + \|x_{1,m}\|^2) \lambda_{\min}^{d-1} > e^D \lambda_{\min}^d$, which, in turn, yields $\|x_{1,m}\|^2 > \lambda_{\min}(e^D - 1) =: C$. Now, if $\|x_{1,j}\|^2 \leq C$, then silo $j$ immediately knows that $\|x_{1,i}\|^2 > C$, where $C$ is a known constant. Since $x_{1,i}$ contains the context information of the user (Alice), this norm condition could immediately reveal that some specific features in the context vector are active (e.g., Alice has both diabetes and heart disease), thus leaking Alice's private and sensitive information to silo $j$.

*Remark* B.1. The above result has two implications: (i) the current proof strategy for Fed-DP guarantee in Dubey & Pentland (2020) does not hold since it essentially relies on the post-processing of DP through silo-level LDP; (ii) Fed-DP could fail to handle reasonable adversary model in cross-silo federated LCBs. That is, even if Algorithm 1 in Dubey & Pentland (2020) satisfies Fed-DP, it still cannot protect Alice's information from being inferred by a malicious silo (which is a typical

---

[3]There is some minor issue in the form of $f$ in Dubey & Pentland (2020). The correct one is given by our restatement of their Algorithm 1, see line 9 in Algorithm 3.

adversary model in cross-silo FL). Thus, we believe that silo-level LDP is a more proper privacy notion for cross-silo federated LCBs.

### B.2 MORE ON VIOLATION OF FED-DP

As shown in the main paper, Algorithm 1 in Dubey & Pentland (2020) also does not satisfy its weaker notion of Fed-DP. To give a more concrete illustration, recall Example 3.1 and let us define $m_{i,j}$ as the message/data sent from silo $i$ to silo $j$ after round $t = 1$. Suppose in the case of Alice, there is no synchronization and hence $m_{i,j} = 0$. On the other hand, in the case of Tracy (i.e., the first user at silo $i$ changes from Alice to Tracy), suppose synchronization is triggered by silo $i$ via rule (1) due to Tracy's data. Then, according to Dubey & Pentland (2020), $m_{i,j} = x_{1,i}y_{1,i} + \mathcal{N}$ (only consider bias vector here), where $\mathcal{N}$ is the injected noise when silo $i$ sends out its data. Now, based on the requirement of Fed-DP, the recommended action at silo $j$ in round $t = 2$ needs to be "similar" or "indistinguishable" in probability under the change from Alice to Tracy. Note that silo $j$ chooses its action at round $t = 2$ based on its local data (which is unchanged) and $m_{i,j}$, via *deterministic* selection rule (i.e., LinUCB) in Algorithm 1 of Dubey & Pentland (2020). Thus, Fed-DP essentially requires $m_{i,j}$ to be close in probability when Alice changes to Tracy, which is definitely not the case (i.e., 0 vs. $x_{1,i}y_{1,i} + \mathcal{N}$). Thus, Algorithm 1 in Dubey & Pentland (2020) also fails Fed-DP.

*Remark B.2.* One can also think from the following perspective: the non-private data-dependent sync rule (i.e., (2)) in Dubey & Pentland (2020) impacts the communicated messages/data as well, which cannot be made private by injecting noise when sending out data. To rescue, a possible approach is to use *private (noisy)* data in rule (2) when determining synchronization (while still injecting noise when sending out data). As a result, whether there exists a synchronization would be "indistinguishable" under Alice or Tracy and hence $m_{i,j}$ now would be similar. However, this approach still suffers the gap in communication cost analysis (see below) and moreover it will incur new challenges in regret analysis, see Appendix D for a detailed discussion on this approach.

### B.3 MORE ON COMMUNICATION COST ANALYSIS

The current analysis in Dubey & Pentland (2020) (cf. Proposition 5) for communication cost (i.e., how many rounds of communication within $T$) essentially follows the approach in the non-private work (Wang et al., 2020) (cf. proof of Theorem 4). However, due to additional privacy noise injected into the communicated data, one key step of the approach in Wang et al. (2020) fails in the private case. In the following, we first point out the issue using notations in Dubey & Pentland (2020).

The key issue in its current proof of Proposition 5 in Dubey & Pentland (2020) is that

$$\log \frac{\det(\mathbf{S}_{i,t+n'})}{\det(\mathbf{S}_{i,t})} > \frac{D}{n'} \tag{3}$$

which appears right above Eq. 4 in Dubey & Pentland (2020) does not hold. More specifically, $[t, t + n']$ is the $i$-th interval between two communication steps and $\mathbf{S}_{i,t}, \mathbf{S}_{i,t+n'}$ are corresponding synchronized private matrices. At the time $t + n'$, we know (2) is satisfied by some silo (say $j \in [M]$), since there is a new synchronization. In the non-private case, $\mathbf{S}_{i,t+n'}$ simply includes some additional local covariance matrices from silos other than $j$, which are positive semi-definite (PSD). As a result, (3) holds. However, in the private case, $\mathbf{S}_{i,t+n'}$ includes the *private* messages from silos other than $j$, which may not be positive semi-definite (PSD), since there are some new covariance matrices as well as *new Gaussian privacy noise* (which could be negative definite). Thus, (3) may not hold anymore.

## C   A GENERIC REGRET ANALYSIS FOR ALGORITHM 1

In this section, we first establish a generic regret bound for Algorithm 1 under sub-Gaussian noise condition, i.e., Lemma C.4. To this end, let us first give the following notations. Fix $B, T \in \mathbb{N}$, we let $K = T/B$ be the total number of communication steps. For all $i \in [M]$ and all $t = kB$, $k \in [K]$, we let $N_{t,i} = \widetilde{W}_{t,i} - \sum_{s=1}^{t} x_{s,i}x_{s,i}^\top$ and $n_{t,i} = \widetilde{U}_{t,i} - \sum_{s=1}^{t} x_{s,i}y_{s,i}$ be the cumulative injected noise up to the $k$-th communication by agent $i$. We further let $H_t := \lambda I_d + \sum_{i \in [M]} N_{t,i}$ and $h_t := \sum_{i \in [M]} n_{t,i}$.

**Assumption C.1** (Regularity). Fix any $\alpha \in (0, 1]$, with probability at least $1 - \alpha$, we have $H_t$ is positive definite and there exist constants $\lambda_{\max}, \lambda_{\min}$ and $\nu$ depending on $\alpha$ such that for all $t = kB$, $k \in [K]$

$$\|H_t\| \leq \lambda_{\max}, \quad \|H_t^{-1}\| \leq 1/\lambda_{\min}, \quad \|h_t\|_{H_t^{-1}} \leq \nu.$$

With the above regularity assumption and the boundedness in Assumption 4.1, we fist establish the following general regret bound of Algorithm 1, which can be viewed as a direct generalization of the results in Shariff & Sheffet (2018); Chowdhury & Zhou (2022b) to the federated case.

**Lemma C.2.** *Let Assumptions C.1 and 4.1 hold. Fix any $\alpha \in (0, 1]$, there exist choices of $\lambda$ and $\{\beta_{t,i}\}_{t \in [T], i \in [M]}$ such that, with probability at least $1 - \alpha$, the group regret of Algorithm 1 satisfies*

$$Reg_M(T) = O\left(\beta_T \sqrt{dMT \log\left(1 + \frac{MT}{d\lambda_{\min}}\right)}\right) + O\left(M \cdot B \cdot d \log\left(1 + \frac{MT}{d\lambda_{\min}}\right)\right),$$

*where $\beta_T := \sqrt{2 \log\left(\frac{2}{\alpha}\right) + d \log\left(1 + \frac{MT}{d\lambda_{\min}}\right)} + \sqrt{\lambda_{\max}} + \nu$.*

Lemma C.4 is a corollary of the above result, which holds by bounding $\lambda_{\max}, \lambda_{\min}, \nu$ under sub-Gaussian privacy noise.

**Assumption C.3** (sub-Gaussian private noise). There exist constants $\widetilde{\sigma}_1$ and $\widetilde{\sigma}_2$ such that for all $t = kB$, $k \in [K]$: (i) $\sum_{i=1}^{M} n_{t,i}$ is a random vector whose entries are independent, mean zero, sub-Gaussian with variance at most $\widetilde{\sigma}_1^2$, and (ii) $\sum_{i=1}^{M} N_{t,i}$ is a random symmetric matrix whose entries on and above the diagonal are independent sub-Gaussian random variables with variance at most $\widetilde{\sigma}_2^2$. Let $\sigma^2 = \max\{\widetilde{\sigma}_1^2, \widetilde{\sigma}_2^2\}$.

Now, we are ready to state Lemma C.4 as follows.

**Lemma C.4** (A generic regret bound of Algorithm 1). *Let Assumptions C.3 and 4.1 hold. Fix time horizon $T \in \mathbb{N}$, batch size $B \in [T]$, confidence level $\alpha \in (0, 1]$. Set $\lambda = \Theta(\max\{1, \sigma(\sqrt{d} + \sqrt{\log(T/(B\alpha))})\})$ and $\beta_{t,i} = \sqrt{2 \log\left(\frac{2}{\alpha}\right) + d \log\left(1 + \frac{Mt}{d\lambda}\right)} + \sqrt{\lambda}$ for all $i \in [M]$. Then, Algorithm 1 achieves group regret*

$$Reg_M(T) = O\left(dMB \log T + d\sqrt{MT} \log(MT/\alpha)\right) + O\left(\sqrt{\sigma MT \log(MT)} d^{3/4} \log^{1/4}(T/(B\alpha))\right)$$

*with probability at least $1 - \alpha$.*

## C.1 PROOFS

*Proof of Lemma C.2.* We divide the proof into the following six steps. Let $\mathcal{E}$ be the event given in Assumption C.1, which holds with probability at least $1 - \alpha$ under Assumption C.1. In the following, we condition on the event $\mathcal{E}$.

**Step 1: Concentration.** In this step, we will show that with high probability, $\left\|\theta^* - \widehat{\theta}_{t,i}\right\|_{V_{t,i}} \leq \beta_{t,i}$ for all $i \in [M]$. Fix an agent $i \in [M]$ and $t \in [T]$, let $t_{\text{last}}$ be the latest communication round of all agents before $t$. By the update rule, we have

$$\widehat{\theta}_{t,i} = V_{t,i}^{-1}(\widetilde{U}_{\text{syn}} + U_i)$$

$$= V_{t,i}^{-1}\left(\sum_{j=1}^{M}\sum_{s=1}^{t_{\text{last}}} x_{s,j} y_{s,j} + \sum_{j=1}^{M} n_{t_{\text{last}},j} + \sum_{s=t_{\text{last}}+1}^{t-1} x_{s,i} y_{s,i}\right)$$

$$= \left(\lambda I + \sum_{j=1}^{M}\sum_{s=1}^{t_{\text{last}}} x_{s,j} x_{s,j}^{\top} + \sum_{j=1}^{M} N_{t_{\text{last}},j} + \sum_{s=t_{\text{last}}+1}^{t-1} x_{s,i} x_{s,i}^{\top}\right)^{-1} \left(\sum_{j=1}^{M}\sum_{s=1}^{t_{\text{last}}} x_{s,j} y_{s,j} + \sum_{j=1}^{M} n_{t_{\text{last}},j} + \sum_{s=t_{\text{last}}+1}^{t-1} x_{s,i} y_{s,i}\right).$$

By the linear reward function $y_{s,j} = \langle x_{s,j}, \theta^* \rangle + \eta_{s,j}$ for all $j \in [M]$ and elementary algebra, we have

$$\theta^* - \widehat{\theta}_{t,i} = V_{t,i}^{-1} \left( H_{t_{\text{last}}} \theta^* - \sum_{j=1}^{M} \sum_{s=1}^{t_{\text{last}}} x_{s,j} \eta_{s,j} - \sum_{s=t_{\text{last}}+1}^{t-1} x_{s,i} \eta_{s,i} - h_{t_{\text{last}}} \right),$$

where we recall that $H_{t_{\text{last}}} = \lambda I + \sum_{j=1}^{M} N_{t_{\text{last}},j}$ and $h_{t_{\text{last}}} = \sum_{j=1}^{M} n_{t_{\text{last}},j}$.

Thus, multiplying both sides by $V_{t,i}^{1/2}$, yields

$$\left\| \theta^* - \widehat{\theta}_{t,i} \right\|_{V_{t,i}} \leq \left\| \sum_{j=1}^{M} \sum_{s=1}^{t_{\text{last}}} x_{s,j} \eta_{s,j} + \sum_{s=t_{\text{last}}+1}^{t-1} x_{s,i} \eta_{s,i} \right\|_{V_{t,i}^{-1}} + \left\| H_{t_{\text{last}}} \theta^* \right\|_{V_{t,i}^{-1}} + \left\| h_{t_{\text{last}}} \right\|_{V_{t,i}^{-1}}$$

$$\overset{(a)}{\leq} \left\| \sum_{j=1}^{M} \sum_{s=1}^{t_{\text{last}}} x_{s,j} \eta_{s,j} + \sum_{s=t_{\text{last}}+1}^{t-1} x_{s,i} \eta_{s,i} \right\|_{(G_{t,i}+\lambda_{\min}I)^{-1}} + \left\| \theta^* \right\|_{H_{t_{\text{last}}}} + \left\| h_{t_{\text{last}}} \right\|_{H_{t_{\text{last}}}^{-1}}$$

$$\overset{(b)}{\leq} \left\| \sum_{j=1}^{M} \sum_{s=1}^{t_{\text{last}}} x_{s,j} \eta_{s,j} + \sum_{s=t_{\text{last}}+1}^{t-1} x_{s,i} \eta_{s,i} \right\|_{(G_{t,i}+\lambda_{\min}I)^{-1}} + \sqrt{\lambda_{\max}} + \nu$$

where (a) holds by $V_{t,i} \succeq H_{t_{\text{last}}}$ and $V_{t,i} \succeq G_{t,i} + \lambda_{\min}I$ with $G_{t,i} := \sum_{j=1}^{M} \sum_{s=1}^{t_{\text{last}}} x_{s,j} x_{s,j}^\top + \sum_{s=t_{\text{last}}+1}^{t-1} x_{s,i} x_{s,i}^\top$ (i.e., non-private Gram matrix) under event $\mathcal{E}$; (b) holds by the boundedness of $\theta^*$ and event $\mathcal{E}$.

For the remaining first term, we can use self-normalized inequality (cf. Theorem 1 in Abbasi-Yadkori et al. (2011)) with a proper filtration[4]. In particular, we have for any $\alpha \in (0,1]$, with probability at least $1 - \alpha$, for all $t \in [T]$

$$\left\| \sum_{j=1}^{M} \sum_{s=1}^{t_{\text{last}}} x_{s,j} \eta_{s,j} + \sum_{s=t_{\text{last}}+1}^{t-1} x_{s,i} \eta_{s,i} \right\|_{(G_{t,i}+\lambda_{\min}I)^{-1}} \leq \sqrt{2 \log \left( \frac{1}{\alpha} \right) + \log \left( \frac{\det(G_{t,i} + \lambda_{\min}I)}{\det(\lambda_{\min}I)} \right)}.$$

Now, using the trace-determinant lemma (cf. Lemma 10 in Abbasi-Yadkori et al. (2011)) and the boundedness condition on $\|x_{s,j}\|$ for all $s \in [T]$ and $j \in [M]$, we have

$$\det(G_{t,i} + \lambda_{\min}I) \leq \left( \lambda_{\min} + \frac{Mt}{d} \right)^d.$$

Putting everything together, we have with probability at least $1 - 2\alpha$, for all $i \in [M]$ and all $t \in [T]$, $\left\| \theta^* - \widehat{\theta}_m \right\|_{V_{t,i}} \leq \beta_{t,i} = \beta_t$, where

$$\beta_t := \sqrt{2 \log \left( \frac{1}{\alpha} \right) + d \log \left( 1 + \frac{Mt}{d\lambda_{\min}} \right)} + \sqrt{\lambda_{\max}} + \nu. \tag{4}$$

**Step 2: Per-step regret.** With the above concentration result, based on our UCB policy for choosing the action, we have the classic bound on the per-step regret $r_{t,i}$, that is, with probability at least $1 - 2\alpha$

$$r_{t,i} = \langle \theta^*, x_{t,i}^* \rangle - \langle \theta^*, x_{t,i} \rangle$$

$$\overset{(a)}{=} \langle \theta^*, x_{t,i}^* \rangle - \text{UCB}_{t,i}(x_{t,i}^*) + \text{UCB}_{t,i}(x_{t,i}^*) - \text{UCB}_{t,i}(x_{t,i}) + \text{UCB}_{t,i}(x_{t,i}) - \langle \theta^*, x_{t,i} \rangle$$

$$\overset{(b)}{\leq} 0 + 0 + 2\beta_{t,i} \|x_{t,i}\|_{V_{t,i}^{-1}} \leq 2\beta_T \|x_{t,i}\|_{V_{t,i}^{-1}}$$

where in (a), we let $\text{UCB}_{t,i}(x) := \langle \widehat{\theta}_{t,i}, x \rangle + \beta_{t,i} \|x\|_{V_{t,i}^{-1}}$; (b) holds by the optimistic fact of UCB (from the concentration), greedy action selection, and the concentration result again.

---

[4] In particular, by the i.i.d noise assumption across time and agents, one can simply construct the filtration sequentially across agents and rounds, which enlarges the single-agent filtration by a factor of $M$.

**Step 3: Regret decomposition by good and bad epochs.** In Algorithm 1, at the end of each synchronization time $t = kB$ for $k \in [K]$, all the agents will communicate with the server by uploading private statistics and downloading the aggregated ones from the server. We then divide time horizon $T$ into epochs by the communication (sync) rounds. In particular, the $k$-th epoch contains rounds between $(t_{k-1}, t_k]$, where $t_k = kB$ is the $k$-th sync round. We define $V_k := \lambda_{\min} I + \sum_{i=1}^{M} \sum_{t=1}^{t_k} x_{t,i} x_{t,i}^\top$, i.e., all the data at the end of the $k$-th communication plus a regularizer. Then, we say that the $k$-th epoch is a "good" epoch if $\frac{\det(V_k)}{\det(V_{k-1})} \leq 2$; otherwise it is a "bad" epoch. Thus, we can divide the group regret into two terms:

$$\mathrm{Reg}_M(T) = \sum_{i \in [M]} \sum_{t \in \text{good epochs}} r_{t,i} + \sum_{i \in [M]} \sum_{t \in \text{bad epochs}} r_{t,i}.$$

**Step 4: Bound the regret in good epochs.** To this end, we introduce an *imaginary* single agent that pulls all the $MT$ actions in the following order: $x_{1,1}, x_{1,2}, \ldots, x_{1,M}, x_{2,1}, \ldots, x_{2,M}, \ldots, x_{T,1}, \ldots, x_{T,M}$. We define a corresponding *imaginary* design matrix $\bar{V}_{t,i} = \lambda_{\min} I + \sum_{p < t, q \in [M]} x_{p,q} x_{p,q}^\top + \sum_{p=t, q<i} x_{p,q} x_{p,q}^\top$, i.e., the design matrix right *before* $x_{t,i}$. The key reason behind this construction is that one can now use the standard result (i.e., the elliptical potential lemma (cf. Lemma 11 in Abbasi-Yadkori et al. (2011))) to bound the summation of bonus terms, i.e., $\sum_{t,i} \|x_{t,i}\|_{\bar{V}_{t,i}^{-1}}$.

Suppose that $t \in [T]$ is within the $k$-th epoch. One key property we will use is that for all $i$, $V_k \succeq \bar{V}_{t,i}$ and $G_{t,i} + \lambda_{\min} I \succeq V_{k-1}$, which simply holds by their definitions. This property enables us to see that for any $t \in$ good epochs, $\det(\bar{V}_{t,i})/\det(G_{t,i} + \lambda_{\min} I) \leq 2$. This is important since by the standard "determinant trick", we have

$$\|x_{t,i}\|_{(G_{t,i} + \lambda_{\min} I)^{-1}} \leq \sqrt{2} \|x_{t,i}\|_{\bar{V}_{t,i}^{-1}}. \tag{5}$$

In particular, this follows from Lemma 12 in Abbasi-Yadkori et al. (2011), that is, for two positive definite matrices $A, B \in \mathbb{R}^{d \times d}$ satisfying $A \succeq B$, then for any $x \in \mathbb{R}^d$, $\|x\|_A \leq \|x\|_B \cdot \sqrt{\det(A)/\det(B)}$. Note that here we also use $\det(A) = 1/\det(A^{-1})$. Hence, we can bound the regret in good epochs as follows.

$$
\begin{aligned}
\sum_{i \in [M]} \sum_{t \in \text{good epochs}} r_{t,i} &\overset{(a)}{\leq} \sum_{i \in [M]} \sum_{t \in \text{good epochs}} \min\{2\beta_T \|x_{t,i}\|_{V_{t,i}^{-1}}, 1\} \\
&\overset{(b)}{\leq} \sum_{i \in [M]} \sum_{t \in \text{good epochs}} \min\{2\beta_T \|x_{t,i}\|_{(G_{t,i} + \lambda_{\min} I)^{-1}}, 1\} \\
&\overset{(c)}{\leq} \sum_{i \in [M]} \sum_{t \in \text{good epochs}} \min\{2\sqrt{2}\beta_T \|x_{t,i}\|_{\bar{V}_{t,i}^{-1}}, 1\} \\
&\overset{(d)}{\leq} \sum_{i \in [M]} \sum_{t \in \text{good epochs}} 2\sqrt{2}\beta_T \min\{\|x_{t,i}\|_{\bar{V}_{t,i}^{-1}}, 1\} \\
&\leq \sum_{i \in [M]} \sum_{t \in [T]} 2\sqrt{2}\beta_T \min\{\|x_{t,i}\|_{\bar{V}_{t,i}^{-1}}, 1\} \\
&\overset{(e)}{\leq} O\left(\beta_T \sqrt{dMT \log\left(1 + \frac{MT}{d\lambda_{\min}}\right)}\right),
\end{aligned} \tag{6}
$$

where (a) holds by the per-step regret bound in Step 2 and the boundedness of reward; (b) follows from the fact that $V_{t,i} \succeq G_{t,i} + \lambda_{\min} I$ under event $\mathcal{E}$; (c) holds by (5) when $t$ is in good epochs; (d) is true since $\beta_T \geq 1$; (e) holds by the elliptical potential lemma (cf. Lemma 11 in Abbasi-Yadkori et al. (2011)).

**Step 5: Bound the regret in bad epochs.** Let $T_{\text{bad}}$ be the total number of rounds in all bad epochs. Thus, the total number of bad rounds across *all* agents are $M \cdot T_{\text{bad}}$. As a result, the cumulative group regret in all these bad rounds are upper bounded by $M \cdot T_{\text{bad}}$ due to the to the boundedness of reward.

We are left to bound $T_{\mathrm{bad}}$. All we need is to bound the $N_{\mathrm{bad}}$ – total number of bad epochs. Then, we have $T_{\mathrm{bad}} = N_{\mathrm{bad}} \cdot B$, where $B$ is the fixed batch size. To this end, recall that $K = T/B$ and define $\Psi := \{k \in [K] : \log \det(V_k) - \log \det(V_{k-1}) > \log 2\}$, i.e., $N_{\mathrm{bad}} = |\Psi|$. Thus, we have

$$\log 2 \cdot |\Psi| \leq \sum_{k \in \Psi} \log \det(V_k) - \log \det(V_{k-1}) \leq \sum_{k \in [K]} \log \det(V_k) - \log \det(V_{k-1})$$

$$\leq d \log \left( 1 + \frac{MT}{d\lambda_{\min}} \right)$$

Hence, we have $N_{\mathrm{bad}} = |\Psi| \leq \frac{d}{\log 2} \log \left( 1 + \frac{MT}{d\lambda_{\min}} \right)$. Thus we can bound the regret in bad epochs as follows.

$$\sum_{i \in [M]} \sum_{t \in \mathrm{bad\ epochs}} r_{t,i} \leq M \cdot T_{\mathrm{bad}} = M \cdot B \cdot N_{\mathrm{bad}} \leq M \cdot B \cdot \frac{d}{\log 2} \log \left( 1 + \frac{MT}{d\lambda_{\min}} \right). \quad (7)$$

**Step 6: Putting everything together.** Now, we substitute the total regret in good epochs given by (6) and total regret in bad epochs given by (7) into the total regret decomposition in Step 3, yields the final cumulative group regret

$$\mathrm{Reg}_M(T) = O\left( \beta_T \sqrt{dMT \log \left( 1 + \frac{MT}{d\lambda_{\min}} \right)} \right) + O\left( M \cdot B \cdot d \log \left( 1 + \frac{MT}{d\lambda_{\min}} \right) \right),$$

where $\beta_T := \sqrt{2 \log \left( \frac{1}{\alpha} \right) + d \log \left( 1 + \frac{MT}{d\lambda_{\min}} \right)} + \sqrt{\lambda_{\max}} + \nu$. Finally, taking a union bound, we have the required result. $\square$

Now, we turn to the proof of Lemma C.4, which is an application of Lemma C.2 we just proved.

*Proof of Lemma C.4.* To prove the result, thanks to Lemma C.2, we only need to determine the three constants $\lambda_{\max}, \lambda_{\min}$ and $\nu$ under the sub-Gaussian private noise assumption in Assumption C.3. To this end, we resort to concentration bounds for sub-Gaussian random vector and random matrix.

To start with, under (i) in Assumption C.3, by the concentration bound for the norm of a vector containing sub-Gaussian entries (cf. Theorem 3.1.1 in Vershynin (2018)) and a union bound over all communication rounds, we have for all $t = kB$ where $k = [T/B]$ and any $\alpha \in (0, 1]$, with probability at least $1 - \alpha/2$, for some absolute constant $c_1$,

$$\left\| \sum_{i=1}^M n_{t,i} \right\| = \|h_t\| \leq \Sigma_n := c_1 \cdot \widetilde{\sigma}_1 \cdot (\sqrt{d} + \sqrt{\log(T/(\alpha B))}).$$

By (ii) in Assumption C.3, the concentration bound for the norm of a sub-Gaussian symmetric random matrix (cf. Corollary 4.4.8 in Vershynin (2018)) and a union bound over all communication rounds, we have for all $t = kB$ where $k = [T/B]$ and any $\alpha \in (0, 1]$, with probability at least $1 - \alpha/2$,

$$\left\| \sum_{i=1}^M N_{t,i} \right\| \leq \Sigma_N := c_2 \cdot \widetilde{\sigma}_2 \cdot (\sqrt{d} + \sqrt{\log(T/(\alpha B))})$$

for some absolute constant $c_2$. Thus, if we choose $\lambda = 2\Sigma_N$, we have $\|H_t\| = \left\| \lambda I_d + \sum_{i=1}^M N_{t,i} \right\| \leq 3\Sigma_N$, i.e., $\lambda_{\max} = 3\Sigma_N$, and $\lambda_{\min} = \Sigma_N$. Finally, to determine $\nu$, we note that

$$\|h_t\|_{H_t^{-1}} \leq \frac{1}{\sqrt{\lambda_{\min}}} \|h_t\| \leq c \cdot \left( \sigma \cdot (\sqrt{d} + \sqrt{\log(T/(\alpha B))}) \right)^{1/2} := \nu,$$

where $\sigma = \max\{\widetilde{\sigma}_1, \widetilde{\sigma}_2\}$. The final regret bound is obtained by plugging the three values into the result given by Lemma C.2. $\square$

## D    DISCUSSION ON PRIVATE ADAPTIVE COMMUNICATION

In the main paper and Appendix B, we have pointed out that the gap in privacy guarantee of Algorithm 1 in Dubey & Pentland (2020) is that its adaptive communication schedule leads to privacy leakage

---

**Algorithm 3** Restatement of Algorithm 1 in (Dubey & Pentland, 2020)

---

1: **Parameters:** Adaptive communication parameter $D$, regularization $\lambda > 0$, confidence radii $\{\beta_{t,i}\}_{t \in [T], i \in [M]}$, feature map $\phi_i : \mathcal{C}_i \times \mathcal{K}_i \to \mathbb{R}^d$, privacy budgets $\varepsilon > 0, \delta \in [0,1]$.

2: **Initialize:** For all $i \in [M]$, $W_i = 0, U_i = 0$, PRIVATIZER with $\varepsilon, \delta$, $\widetilde{W}_{\text{syn}} = 0$, $\widetilde{U}_{\text{syn}} = 0$,

3: **for** $t = 1, \ldots, T$ **do**

4:     **for** each agent $i = 1, \ldots, M$ **do**

5:         $V_{t,i} = \lambda I + \widetilde{W}_{\text{syn}} + W_i$, $\widehat{\theta}_{t,i} = V_{t,i}^{-1}(\widetilde{U}_{\text{syn}} + U_i)$

6:         Play arm $a_{t,i} = \operatorname{argmax}_{a \in \mathcal{K}_i} \langle \phi_i(c_{t,i}, a), \widehat{\theta}_{t,i} \rangle + \beta_{t,i} \| \phi_i(c_{t,i}, a) \|_{V_{t,i}^{-1}}$ and set $x_{t,i} = \phi_i(c_{t,i}, a_{t,i})$

7:         Observe reward $y_{t,i}$

8:         Update $W_i = W_i + x_{t,i} x_{t,i}^\top$, $U_i = U_i + x_{t,i} y_{t,i}$

9:         **if** $\log \det(V_{t,i} + x_{t,i} x_{t,i}^\top) - \log \det(V_{\text{last}}) > \frac{D}{t - t_{\text{last}}}$ **then**

10:            Send a signal to the server to start a synchronization round.

11:         **end if**

12:         **if** a synchronization is started **then**

13:            Send $W_i$ and $U_i$ to PRIVATIZER

14:            PRIVATIZER sends private cumulative statistics $\widetilde{W}_{t,i}, \widetilde{U}_{t,i}$ to server

15:            Server aggregates $\widetilde{W}_{\text{syn}} = \widetilde{W}_{\text{syn}} + \sum_{j=1}^{M} \widetilde{W}_{t,j}$ and $\widetilde{U}_{\text{syn}} = \widetilde{U}_{\text{syn}} + \sum_{j=1}^{M} \widetilde{U}_{t,j}$

16:            Receive $\widetilde{W}_{\text{syn}}$ and $\widetilde{U}_{\text{syn}}$ from the server

17:            Reset $W_i = 0, U_i = 0, t_{\text{last}} = t$ and $V_{\text{last}} = \lambda I + \widetilde{W}_{syn}$

18:         **end if**

19:     **end for**

20: **end for**

---

due to its dependence on non-private data. As mentioned in Remark B.1, one possible approach is to use private data to determine the sync in (2). This will resolve the privacy issue. However, the same issue in communication cost still remains (due to privacy noise), and hence $O(\log T)$ communication does not hold. Moreover, this new approach will also lead to new challenges in regret analysis, when compared with its current one in Dubey & Pentland (2020) and the standard one in Wang et al. (2020).

To better illustrate the new challenges, let us restate Algorithm 1 in Dubey & Pentland (2020) using our notations and first focus on how to establish the regret based on its current adaptive schedule (which has the issue of privacy leakage). After we have a better understanding of the idea, we will see how new challenges come up when one uses private data for an adaptive schedule.

As shown in Algorithm 3, the key difference compared with our fixed-batch schedule is highlighted in color. Note that we only focus on silo-level LDP and use PRIVATIZER to represent a general protocol that can privatize the communicated data (e.g., $\mathcal{P}$ or the standard tree-based algorithm in Dubey & Pentland (2020)).

## D.1 REGRET ANALYSIS UNDER NON-PRIVATE ADAPTIVE SCHEDULE

In this section, we demonstrate the key step in establishing the regret with the non-private adaptive communication schedule in Algorithm 3 (i.e., line 9). It turns out that the regret analysis is very similar to our proof for Lemma C.2 for the fixed batch case, in that the only key difference lies in Step 5 when bounding the regret in bad epochs[5]. The main idea behind adaptive communication is: *whenever the accumulated local regret at any agent exceeds a threshold, then synchronization is required to keep the data homogeneous among agents.* This idea is directly reflected in the following analysis.

---

[5]There is another subtle but important difference, which lies in the construction of filtration that is required to apply the standard self-normalized inequality to establish the concentration result. We believe that one cannot directly use the standard filtration (e.g., Abbasi-Yadkori et al. (2011)) in the adaptive case, and hence more care is indeed required.

**Bound the regret in bad epochs (adaptive communication case).** Let's consider an arbitrary bad epoch $k$, i.e., $(t_{k-1}, t_k]$, where $t_k$ is the round for the $k$-th communication. For all $i$, we want to bound the total regret between $(t_{k-1}, t_k]$, denoted by $R_i^k$. That is, the local regret between any two communications (in the bad epoch) will not be too large. For now, suppose we already have such a bound $U$ (which will be achieved by adaptive communication later), i.e., $R_i^k \le U$ for all $i, k$, we can easily bound the total regret in bad epochs. To see this, recall that $\Psi := \{k \in [K] : \log \det(V_k) - \log \det(V_{k-1}) > \log 2\}$, i.e., $N_{bad} = |\Psi|$, we have

$$\sum_i \sum_{t \in \text{bad epochs}} r_{t,i} = \sum_i \sum_{k \in \Psi} R_i^k = O\left(|\Psi| MU\right).$$

Plugging in $N_{bad} = |\Psi| \le \frac{d}{\log 2} \log\left(1 + \frac{MT}{d\lambda}\right)$, we have the total regret for bad epochs. Now, we are only left to find $U$. Here is the place where the adaptive schedule in the algorithm comes in. First, note that

$$\sum_{t_{k-1} < t < t_k} r_{t,i} \overset{(a)}{\le} \sum_{t_{k-1} < t < t_k} \min\{2\beta_T \|x_{t,i}\|_{V_{t,i}^{-1}}, 1\} \tag{8}$$

$$\overset{(b)}{\le} O\left(\beta_T \sqrt{(t_k - t_{k-1}) \log \frac{\det V_{t_k,i}}{\det V_{\text{last}}}}\right) \tag{9}$$

$$\overset{(c)}{\le} O\left(\beta_T \sqrt{D}\right),$$

where (a) holds by boundedness of reward; (b) follows from the elliptical potential lemma, i.e., $V_{\text{last}}$ is PSD under event $\mathcal{E}$ and $V_{t,i} = V_{t-1,i} + x_{t-1,i} x_{t-1,i}^\top$ for all $t \in (t_{k-1}, t_k)$; (c) holds by the adaptive schedule in line 9 of Algorithm 3. As a result, we have $R_i^k \le O\left(\beta_T \sqrt{D}\right) + 1$, where the regret at round $t_k$ is at most 1 by the boundedness of reward. With a proper choice of $D$, one can obtain a final regret bound.

## D.2 Challenges in Regret Analysis under Private Adaptive Schedule

Now, we will discuss new challenges when one uses private data for an adaptive communication schedule. In this case, one needs to first privatize the new local gram matrices (e.g., $\sum_{s=t_{\text{last}}+1}^t x_{s,i} x_{s,i}^\top$) before being used in the determinant condition. This can be done by using standard tree-based algorithm with each data point as $x_{s,i} x_{s,i}^\top$. With this additional step, now the determinant condition becomes

$$\log \det(\widetilde{V}_{t,i}) - \log \det(V_{\text{last}}) > \frac{D}{t - t_{\text{last}}}, \tag{10}$$

where $\widetilde{V}_{t,i} := V_{\text{last}} + \sum_{s=t_{\text{last}}+1}^t x_{s,i} x_{s,i}^\top + N_{t,i}^{\text{loc}}$ and $N_{t,i}^{\text{loc}}$ is the new local injected noise for private schedule up to time $t$. Now suppose one uses (10) to determine $t_k$. Then, it does not imply that (9) is upper bounded by $\beta_T \sqrt{D}$. That is, $\frac{\det(\widetilde{V}_{t,i})}{\det(V_{\text{last}})} \le D'$ does not necessarily mean that $\frac{\det(V_{\text{last}} + \sum_{s=t_{\text{last}}+1}^t x_{s,i} x_{s,i}^\top)}{\det(V_{\text{last}})} \le D'$.

One may try to work around (8) by first using $G_{t,i} + \lambda_{\min} I$ to lower bound $V_{t,i}$. Then, (9) becomes $O\left(\beta_T \sqrt{(t_k - t_{k-1}) \log \frac{\det(G_{t_k,i} + \lambda_{\min} I)}{\det(G_{t_{k-1},i} + \lambda_{\min} I)}}\right)$, which again cannot be bouded based on the rule given by (10). To see this, note that $\frac{\det(\widetilde{V}_{t_k-1,i})}{\det(V_{\text{last}})} \le D'$ only implies that $\frac{\det(G_{t_k,i} + \lambda_{\min} I)}{\det(G_{t_{k-1},i} + \lambda_{\max} I)} \le D'$.

## E Additional Details on Federated LCBs under Silo-Level LDP

In this section, we provide details for Section 5.1. In particular, we present the proof for Theorem 5.1 and the alternative privacy protocol for silo-level LDP.

### E.1 Proof of Theorem 5.1

*Proof of Theorem 5.1.* **Privacy.** We only need to show that $\mathcal{P}$ in Algorithm 2 with a proper choice of $\sigma_0$ satisfies $(\varepsilon, \delta)$-DP for all $k \in [K]$, which implies that the full transcript of the communication is private in Algorithm 1 for any local agent $i$.

First, we recall that the (multi-variate) Gaussian mechanism satisfies zero-concentrated differential privacy (zCDP) (Bun & Steinke, 2016). In particular, by Bun & Steinke (2016, Lemma 2.5), we have that computation of each node (p-sum) in the tree is $\rho$-zCDP with $\rho = \frac{L^2}{2\sigma_0^2}$. Then, from the construction of the binary tree in $\mathcal{P}$, one can easily see that one single data point $\gamma_i$ (for all $i \in [K]$) only impacts at most $1 + \log(K)$ nodes. Thus, by *adaptive* composition of zCDP (cf. Lemma 2.3 in Bun & Steinke (2016)), we have that the entire releasing of all p-sums is $(1 + \log K)\rho$-zCDP. Finally, we will use the conversion lemma from zCDP to approximated DP (cf. Proposition 1.3 in Bun & Steinke (2016)). In particular, we have that $\rho_0$-zCDP implies $(\varepsilon = \rho_0 + 2\sqrt{\rho_0 \cdot \log(1/\delta)}, \delta)$-DP for all $\delta > 0$. In other words, to achieve a given $(\varepsilon, \delta)$-DP, it suffices to achieve $\rho_0$-zCDP with $\rho_0 = f(\varepsilon, \delta) := (\sqrt{\log(1/\delta) + \varepsilon} - \sqrt{\log(1/\delta)})^2$. In our case, we have $\rho_0 = (1 + \log(K))\rho = (1 + \log(K))\frac{L^2}{2\sigma_0^2}$. Thus, we have $\sigma_0^2 = (1 + \log(K))\frac{L^2}{2\rho_0} = (1 + \log(K))\frac{L^2}{2f(\varepsilon, \delta)}$. To simply it, one can lower bound $f(\varepsilon, \delta)$ by $\frac{\varepsilon^2}{4\log(1/\delta) + 4\varepsilon}$ (cf. Remark 15 in Steinke (2022)). Therefore, to obtain $(\varepsilon, \delta)$-DP, it suffices to set $\sigma_0^2 = 2 \cdot L^2 \cdot \frac{(1 + \log(K))(\log(1/\delta) + \varepsilon)}{\varepsilon^2}$. Note that there are two streams of data in Algorithm 1, and hence it suffices to ensure that each of them is $(\varepsilon/2, \delta/2)$-DP. This gives us the final noise level $\sigma_0^2 = 8\frac{(1 + \log(K))(\log(2/\delta) + \varepsilon)}{\varepsilon^2}$ (note that by boundedness assumption $L = 1$ in our case).

**Regret.** In order to establish the regret bound, thanks to Lemma C.4, we only need to determine the maximum noise level in the learning process. Recall that $\sigma_0^2 = 8 \cdot \frac{(1 + \log(K))(\log(2/\delta) + \varepsilon)}{\varepsilon^2}$ is the noise level for both streams (i.e., $\gamma^{\text{bias}}$ and $\gamma^{\text{cov}}$). Now, by the construction of binary tree in $\mathcal{P}$, one can see that each prefix sum $\sum[1, k]$ only involves at most $1 + \log(k)$ tree nodes. Thus, we have that the noise level in $n_{t,i}$ and $N_{t,i}$ are upper bounded by $(1 + \log(K))\sigma_0^2$. As a result, the overall noise level across all $M$ silos is upper bounded by $\sigma_{\text{total}}^2 = M(1 + \log(K))\sigma_0^2$. Finally, setting $\sigma^2$ in Lemma C.4 to be the noise level $\sigma_{\text{total}}^2$, yields the required result. $\quad\square$

### E.2 Alternative Privacy Protocol for Silo-Level LDP

For silo-level LDP, each local randomizer can simply be the standard tree-based algorithm, i.e., releasing the prefix sum at each communication step $k$ (rather than p-sum in Algorithm 2). The analyzer now becomes a simple aggregation. As before, no shuffler is required in this case. This alternative protocol is given by Algorithm 4, which is essentially the main protocol used in Dubey & Pentland (2020).

It can be seen that both privacy and regret guarantees under this $\mathcal{P}_{\text{alt}}$ are the same as Theorem 5.1. To see this, for privacy, the prefix sum is a post-processing of the p-sums. Thus, since we have already shown that the entire releasing of p-sums is private in the proof of Theorem 5.1, hence the same as the prefix sum. Meanwhile, the total noise level at the server is the same as before. Thus, by Lemma C.4, we have the same regret bound.

## F Additional Details on Federated LCBs under SDP

In this section, we provide more detailed discussions on SDP and present the proof for Theorem 5.3 (SDP via amplification lemma) and Theorem 5.5 (SDP via vector sum).

First, let us start with some general discussions.

**Importance of communicating P-sums.** For SDP, it is important to communicate P-sums rather than prefix sum. Note that communicating noisy p-sums in our privacy protocol $\mathcal{P}$ rather than the noisy prefix sum (i.e., the sum from beginning as done in Dubey & Pentland (2020)) plays a key role in achieving optimal regret with shuffling. To see this, both approaches can guarantee silo-level LDP. By our new amplification lemma, privacy guarantee can be amplified by $1/\sqrt{M}$ in $\varepsilon$ for each

---

**Algorithm 4** $\mathcal{P}_{\text{alt}}$, an alternative privacy protocol for silo-level LDP

---

1: **Procedure:** Local Randomizer $\mathcal{R}$
2: `// Input: stream data` $\gamma = (\gamma_i)_{i \in [K]}$`; privacy parameters` $\varepsilon, \delta$`; Output: private prefix sum`
3:    **for** $k = 1, \dots, K$ **do**
4:       Express $k$ in binary form: $k = \sum_j \text{Bin}_j(k) \cdot 2^j$
5:       Find the index of first one $i_k := \min\{j : \text{Bin}_j(k) = 1\}$
6:       Compute p-sum $\alpha_{i_k} = \sum_{j < i} \alpha_j + \gamma_k$.
7:       Add noise to p-sum $\widehat{\alpha}_{i_k} = \alpha_{i_k} + \mathcal{N}(0, \sigma_0^2 I)$
8:       Output private prefix sum $\widetilde{s}_k = \sum_{j : \text{Bin}_j(k) = 1} \widehat{\alpha}_j$
9:    **end for**
10: **end procedure**
11: **Procedure: Analyzer** $\mathcal{A}$
12: `// Input: a collection of` $M$ `data points,` $y = \{y_i\}_{i \in [M]}$`; Output: Aggregated sum`
13:    Output $\widetilde{y} = \sum_{i \in [M]} y_i$
14: **end procedure**

---

of the $K$ shuffled outputs, where $K = T/B$ is total communication rounds. Now, if the prefix sum is released to the shuffler, then any single data point participates in at most $K$ shuffle mechanisms, which would blow up $\varepsilon$ by a factor of $O(\sqrt{K})$ (by advanced composition (Dwork & Roth, 2014)). This would eventually lead to a $K^{1/4}$ factor blow up in regret due to privacy. Similarly, if we apply $\mathcal{P}_{\text{Vec}}$ to the data points in the prefix sum, then again a single data point can participate in at most $K$ shuffled outputs.

On the other hand, if only noisy p-sums are released for shuffling at each communication round $k \in [K]$ (as in our protocol $\mathcal{P}$) or only the data points in each p-sum are used in $\mathcal{P}_{\text{Vec}}$ (as in our protocol in $\mathcal{P}_{\text{Vec}}^{\mathcal{T}}$), then due to the binary-tree structure, each data point only participates in at most $\log K$ shuffled mechanisms, which only leads to $O(\sqrt{\log K})$ blow-up of $\varepsilon$; hence allowing us to achieve the desired $\widetilde{O}(\sqrt{MT})$ regret scaling, and close the gap present under silo-level LDP.

*Remark* F.1 (Shuffled tree-based mechanism). Both the protocol $\mathcal{P}$ in Algorithm 2 along with our new amplification lemma and protocol $\mathcal{P}_{\text{Vec}}^{\mathcal{T}}$ in Algorithm 5 can be treated as a black-box method, which integrates shuffling into the tree-based mechanism while providing formal guarantees for continual release of sum statistics. Hence, it can be applied to other federated online learning problems beyond contextual bandits.

### F.1 AMPLIFICATION LEMMA FOR SDP

We first formally introduce our new amplification lemma, which is the key to our analysis, as mentioned in the main paper.

The motivation for our new amplification result is two-fold: (i) Existing results on privacy amplification via shuffling (e.g., Feldman et al. (2022); Erlingsson et al. (2019); Cheu et al. (2019); Balle et al. (2019)) are only limited to the standard LDP case, i.e., each local dataset has size $n = 1$, which is not applicable in our case where each silo runs a DP (rather than LDP) mechanism over a dataset of size $n = T$; (ii) Although a recent work (Lowy & Razaviyayn, 2021) establishes a general amplification result for the case of $n > 1$, it introduces a very large value for the final $\delta$ that scales linearly with $n$ due to group privacy.

We first present the key intuition behind our new lemma. Essentially, as in Lowy & Razaviyayn (2021), we follow the nice idea of hiding among the clones introduced in Feldman et al. (2022). That is, the output from silo 2 to $n$ can be similar to that of silo 1 by the property of DP (i.e., creating clones). The key difference between $n = 1$ and $n > 1$ is that in the latter case, the similarity distance between the output of silo 1 and $j$ ($j > 1$) will be larger as in this case all $n > 1$ data points among two silos could be different. To capture this, Lowy & Razaviyayn (2021) resorts to group privacy for

general DP local randomizers.[6] However, group privacy for approximate DP will introduce a large value for $\delta$. Thus, since we know that each local randomizer in our case is the Gaussian mechanism, we can capture the similarity of outputs between silo 1 and $j$ ($j > 1$) by directly bounding the sensitivity. This helps to avoid the large value for the final $\delta$. Specifically, we have the following result, which can be viewed as a refinement of Theorem D.5 in Lowy & Razaviyayn (2021) when specified to the Gaussian mechanism. We follow the notations in Lowy & Razaviyayn (2021) for easy comparison.

**Lemma F.2** (Amplification lemma for Gaussian mechanism). *Let* $\mathbf{X} = (X_1, \cdots, X_N) \in \mathcal{X}^{N \times n}$ *be a distributed data set, i.e.,* $N$ *silos each with* $n$ *data points. Let* $r \in \mathbb{N}$ *and let* $\mathcal{R}_r^{(i)}(\mathbf{Z}, \cdot) : \mathcal{X}^n \to \mathcal{Z} := \mathbb{R}^d$ *be a Gaussian mechanism with* $(\varepsilon_0^r, \delta_0^r)$-*DP,* $\varepsilon_0^r \in (0, 1)$[7]*, for all* $\mathbf{Z} = Z_{(1:r-1)}^{(1:N)} \in \mathcal{Z}^{(r-1) \times N}$ *and* $i \in [N]$*, where* $\mathcal{X}$ *is an arbitrary set. Suppose for all* $i$*,* $\max_{any\,pair(X,X')} \left\| \mathcal{R}_r^{(i)}(\mathbf{Z}, X) - \mathcal{R}_r^{(i)}(\mathbf{Z}, X') \right\| \leq n \cdot \max_{adjacent\,pair(X,X')} \left\| \mathcal{R}_r^{(i)}(\mathbf{Z}, X) - \mathcal{R}_r^{(i)}(\mathbf{Z}, X') \right\|$.[8] *Given* $\mathbf{Z} = Z_{(1:r-1)}^{(1:N)}$*, consider the shuffled algorithm* $\mathcal{A}_s^r : \mathcal{X}^{n \times N} \times \mathcal{Z}^{(r-1) \times N} \to \mathcal{Z}^N$ *that first samples a random permutation* $\pi$ *of* $[N]$ *and then computes* $Z_r = (Z_r^{(1)}, \cdots, Z_r^{(N)})$*, where* $Z_r^{(i)} := \mathcal{R}_r^{(i)}(\mathbf{Z}, X_{\pi(i)})$*. Then, for any* $\delta \in [0,1]$ *such that* $\varepsilon_0^r \leq \frac{1}{n} \ln \left( \frac{N}{16 \log(2/\delta)} \right)$*,* $\mathcal{A}_s^r$ *is* $(\varepsilon^r, \delta^r)$-*DP, where*

$$\varepsilon^r := \ln \left[ 1 + \left( \frac{e^{\varepsilon_0^r} - 1}{e^{\varepsilon_0^r} + 1} \right) \left( \frac{8\sqrt{e^{n\varepsilon_0^r} \log(4/\delta)}}{\sqrt{N}} + \frac{8e^{n\varepsilon_0^r}}{N} \right) \right]$$

$$\delta^r := \left( \frac{e^{\varepsilon_0^r} - 1}{e^{\varepsilon_0^r} + 1} \right) \delta + N(e^{\varepsilon^r} + 1)(1 + e^{-\varepsilon_0^r}/2)\delta_0^r.$$

*If* $\varepsilon_0^r \leq 1/n$*, choosing* $\delta = Nn\delta_0^r$ *yields* $\varepsilon^r = O\left( \frac{\varepsilon_0^r \sqrt{\log(1/(nN\delta_0^r))}}{\sqrt{N}} \right)$ *and* $\delta^r = O(N\delta_0^r)$*, where* $\delta_0^r \leq 1/(Nn)$*.*

### F.2 VECTOR SUM PROTOCOL FOR SDP

One limitation of our first scheme for SDP is that the privacy guarantee holds only for very small values of $\varepsilon$. This comes from two factors: one is due to the fact that standard $1/\sqrt{M}$ amplification result requires the local privacy budget to be close to one; the other one comes from the fact that now the local dataset could be $n = T$, which further reduces the range of valid $\varepsilon$.

In this section, we give the vector sum protocol in Cheu et al. (2021) for easy reference. Let's also give a concrete example to illustrate how to combine Algorithm 6 with Algorithm 5. Consider a fixed $k = 6$. Then, for each agent, we have $\alpha_{i_6} = \gamma_5 + \gamma_6$. That is, consider the case of summing bias vectors, for agent $i \in [M]$, $\gamma_5 = \sum_{t=4B+1}^{5B} x_{t,i}y_{t,i}$ and $\gamma_6 = \sum_{t=5B+1}^{6B} x_{t,i}y_{t,i}$. Then, $\mathcal{D}_6$ consists of $2B$ data points, each of which is a single bias vector. Now, $\mathcal{R}_{\text{vec}}$ and $\mathcal{A}_{\text{vec}}$ (as well the shuffler) work together to compute the noisy sum of $2B \cdot M$ data points. In particular, denote by $\mathcal{P}_{\text{vec}}$ the whole process, then we have $\widetilde{\alpha}_{i_6} = \mathcal{P}_{\text{vec}}(\mathcal{D}_6^M)$, where $\mathcal{D}_6^M$ is the data set that consists of $n = 2B \cdot M$ data points, each of them is a single bias vector.

Next, we present more details on the implementations, i.e., the parameter choices of $g, b, p$. Let's consider $k = 6$ again as an example. In this case, the total number of data points that participate in $\mathcal{P}_{\text{vec}}$ is $n = 2B \cdot M$. Then, according to the proof of Theorem C.1 in Chowdhury & Zhou (2022b),

---

[6]This is because it mainly focuses on the lower bound, where one needs to be general to handle any mechanisms.

[7]Note that standard Gaussian mechanism only applies to the regime when $\varepsilon < 1$. In our case, $\varepsilon_0^r$ is often less than 1. Gaussian mechanism also works for the regime $\varepsilon > 1$, in this case, $\sigma^2 \approx 1/\varepsilon$ rather than $1/\varepsilon^2$. With minor adjustment of the final $\varepsilon^r$, our proof can be extended.

[8]This is w.l.o.g; one can easily generalize it to any upper bound that is a function of $n$.

[9]In our application, each data point means a bias vector or a covariance matrix. See Appendix F.2 for a concrete example.

---

**Algorithm 5** $\mathcal{P}_{\text{Vec}}^{\mathcal{T}}$, another privacy protocol used in Algorithm 1

---

1: **Procedure:** Local Randomizer $\mathcal{R}$ at each agent
2: // Input: stream data $(\gamma_1, \ldots, \gamma_K)$, privacy budgets $\varepsilon > 0, \delta \in (0, 1]$
3:     **for** $k = 1, \ldots, K$ **do**
4:         Express $k$ in binary form: $k = \sum_j \text{Bin}_j(k) \cdot 2^j$
5:         Find index of first one $i_k = \min\{j : \text{Bin}_j(k) = 1\}$
6:         Let $\mathcal{D}_k$ be the set of all data points[9] that contribute to $\alpha_{i_k} = \sum_{j < i_k} \alpha_j + \gamma_k$
7:         Output $y_k = \mathcal{R}_{\text{Vec}}(\mathcal{D}_k)$ // apply $R_{\text{Vec}}$ in Algorithm 6 to each data point
8:     **end for**
9: **Procedure:** Analyzer $\mathcal{A}$ at server
10: // Input: stream data from $\mathcal{S}$: $\{\bar{y}_k = (\bar{y}_{k,1}, \ldots, \bar{y}_{k,M})\}_{k \in [K]}$
11:     **for** $k = 1, \ldots, K$ **do**
12:         Express $k$ in binary and find index of first one $i_k$
13:         Add all messages from $M$ agents: $\widetilde{\alpha}_{i_k} = \mathcal{A}_{\text{Vec}}(\bar{y}_k)$ // apply $A_{\text{Vec}}$ in Algorithm 6
14:         Output: $\widetilde{s}_k = \sum_{j : \text{Bin}_j(k) = 1} \widetilde{\alpha}_j$
15:     **end for**

---

we have

$$g = \max\{2\sqrt{n}, d, 4\}, \quad b = \frac{24 \cdot 10^4 \cdot g^2 \cdot \left(\log\left(\frac{4 \cdot (d^2 + 1)}{\delta}\right)\right)^2}{\varepsilon^2 n}, \quad p = 1/4.$$

---

**Algorithm 6** $P_{\text{vec}}$, a shuffle protocol for vector summation (Cheu et al., 2021)

---

1: **Input:** Database of $d$-dimensional vectors $\mathbf{X} = (\mathbf{x}_1, \cdots, \mathbf{x}_n)$; privacy parameters $\varepsilon, \delta$; $L$.
2: **procedure:** Local Randomizer $R_{\text{vec}}(\mathbf{x}_i)$
3:     **for** $j \in [d]$ **do**
4:         Shift component to enforce non-negativity: $\mathbf{w}_{i,j} \leftarrow \mathbf{x}_{i,j} + L$
5:         $\mathbf{m}_j \leftarrow \mathcal{R}_{1D}(\mathbf{w}_{i,j})$
6:     **end for**
7:     Output labeled messages $\{(j, \mathbf{m}_j)\}_{j \in [d]}$
8: **end procedure**
9: **procedure: Analyzer** $A_{\text{vec}}(\mathbf{y})$
10:     **for** $j \in [d]$ **do**
11:         Run analyzer on coordinate $j$'s messages $z_j \leftarrow \mathcal{A}_{1D}(\mathbf{y}_j)$
12:         Re-center: $o_j \leftarrow z_j - n \cdot L$
13:     **end for**
14:     Output the vector of estimates $\mathbf{o} = (o_1, \cdots o_d)$
15: **end procedure**

---

**Algorithm 7** $\mathcal{P}_{1D}$, a shuffle protocol for summing scalars (Cheu et al., 2021)

---

1: **Input:** Scalar database $X = (x_1, \cdots x_n) \in [0, L]^n$; $g, b \in \mathbb{N}$; $p \in (0, \frac{1}{2})$.
2: **procedure: Local Randomizer** $\mathcal{R}_{1D}(x_i)$
3:     $\bar{x}_i \leftarrow \lfloor x_i g / L \rfloor$.
4:     Sample rounding value $\eta_1 \sim \mathbf{Ber}(x_i g / L - \bar{x}_i)$.
5:     Set $\widehat{x}_i \leftarrow \bar{x}_i + \eta_1$.
6:     Sample privacy noise value $\eta_2 \sim \mathbf{Bin}(b, p)$.
7:     Report $\mathbf{y}_i \in \{0, 1\}^{g+b}$ containing $\widehat{x}_i + \eta_2$ copies of 1 and $g + b - (\widehat{x}_i + \eta_2)$ copies of 0.
8: **end procedure**
9: **procedure: Analyzer** $\mathcal{A}_{1D}(\mathcal{S}(\mathbf{y}_1, \ldots, \mathbf{y}_n))$
10:     Output estimator $\frac{L}{g}((\sum_{i=1}^n \sum_{j=1}^{b+g} (\mathbf{y}_i)_j) - pbn)$.
11: **end procedure**

---

## F.3 PROOFS

First, we present proof of Theorem 5.3.

*Proof of Theorem 5.3.* **Privacy.** In this proof, we directly work on approximate DP. By the boundedness assumption and Gaussian mechanism, we have that with $\sigma_0^2 = \frac{2L^2 \log(1.25/\widehat{\delta}_0)}{\widehat{\varepsilon}_0^2}$, $\mathcal{R}$ in $\mathcal{P}$ is $(\widehat{\varepsilon}_0, \widehat{\delta}_0)$-DP for each communication round $k \in [K]$ (provided $\widehat{\varepsilon}_0 \leq 1$). Now, by our amplification lemma (Lemma F.2), we have that the shuffled output is $(\widehat{\varepsilon}, \widehat{\delta})$-DP with $\widehat{\varepsilon} = O\left( \frac{\widehat{\varepsilon}_0 \sqrt{\log(1/(TM\widehat{\delta}_0))}}{\sqrt{M}} \right)$ and $\widehat{\delta} = O(M\widehat{\delta}_0)$ (provided $\widehat{\varepsilon}_0 \leq 1/T$ and $\widehat{\delta}_0 \leq 1/(MT)$). Here we note that in our case, $N = M$ and $n = T$, where $n = T$ follows from the fact that there exists $\alpha_i$ in the tree that corresponds to the sum of $T$ data points. Moreover, since the same mechanism is run at all silos, shuffling-then-privatizing is the same as first privatizing-then-shuffling the outputs. Next, we apply the advanced composition theorem (cf. Theorem 3.20 in Dwork & Roth (2014)). In particular, by the binary tree structure, each data point involves only $\kappa := 1 + \log(K)$ times in the output of $\mathcal{R}$. Thus, to achieve $(\varepsilon, \delta)$-DP, it suffices to have $\widehat{\varepsilon} = \frac{\varepsilon}{2\sqrt{2\kappa \log(2/\delta)}}$ and $\widehat{\delta} = \frac{\delta}{2\kappa}$. Using all these equations, we can solve for $\widehat{\varepsilon}_0 = C_1 \cdot \frac{\varepsilon \sqrt{M}}{\sqrt{\kappa \log(1/\delta) \log(\kappa/(\delta T))}}$ and $\widehat{\delta}_0 = C_2 \cdot \frac{\delta}{M\kappa}$, for some constants $C_1 > 0$ and $C_2 > 0$. To satisfy the conditions on $\widehat{\varepsilon}_0$ and $\widehat{\delta}_0$, we have $\varepsilon \leq \frac{\sqrt{\kappa}}{C_1 T \sqrt{M}}$ and $\delta \leq \frac{\kappa}{C_2 T}$. With the choice of $\widehat{\varepsilon}_0$ and $\widehat{\delta}_0$, we have the noise variance $\sigma_0^2 = O\left( \frac{2L^2 \beta \log(1/\delta) \log(\kappa/(\delta T)) \log(M\kappa/\delta)}{\varepsilon^2 M} \right)$. Thus, we can apply $\mathcal{P}$ to the bias and covariance terms (with $L = 1$), respectively.

**Regret.** Again, we simply resort to our Lemma C.4 for the regret analysis. In particular, we only need to determine the maximum noise level in the learning process. Note that $\sigma_0^2 = O\left( \frac{2L^2 \kappa \log(1/\delta) \log(\kappa/(\delta T)) \log(M\kappa/\delta)}{\varepsilon^2 M} \right)$ is the noise level injected for both bias and covariance terms. Now, by the construction of the binary tree in $\mathcal{P}$, one can see that each prefix sum only involves at most $1 + \log(k)$ tree nodes. As a result, the overall noise level across all $M$ silos is upper bounded by $\sigma_{\text{total}}^2 = M\kappa\sigma_0^2$. Finally, setting $\sigma^2$ in Lemma C.4 to be the noise level $\sigma_{\text{total}}^2$, yields the required result. □

Now, we prove Theorem 5.5.

*Proof of Theorem 5.5.* **Privacy.** For each calculation of the noisy synchronized p-sum, there exist parameters for $\mathcal{P}_{\text{Vec}}$ such that it satisfies $(\varepsilon_0, \delta_0)$-SDP where $\varepsilon_0 \in (0, 15]$ and $\delta_0 \in (0, 1/2)$ (see Lemma 3.1 in Cheu et al. (2021) or Theorem 3.5 in Chowdhury & Zhou (2022b)). Then, by the binary tree structure, each single data point (bias vector or covariance matrix) only participates in at most $\kappa := 1 + \log(K)$ runs of $\mathcal{P}_{\text{Vec}}$. Thus, to achieve $(\varepsilon, \delta)$-DP, it suffices to have $\varepsilon_0 = \frac{\varepsilon}{2\sqrt{2\kappa \log(2/\delta)}}$ and $\delta_0 = \frac{\delta}{2\kappa}$ by advanced composition theorem. Thus, for any $\varepsilon \in (0, 30\sqrt{2\kappa \log(2/\delta)})$ and $\delta \in (0, 1)$, there exist parameters for $\mathcal{P}_{\text{Vec}}$ such that the entire calculations of noisy p-sums are $(\varepsilon, \delta)$-SDP. Since we have two streams of data (bias and covariance), we finally have that for any $\varepsilon \in (0, 60\sqrt{2\kappa \log(2/\delta)})$ and $\delta \in (0, 1)$, there exist parameters for $\mathcal{P}_{\text{Vec}}$ such that Algorithm 1 with $\mathcal{P}_{\text{Vec}}^{\mathcal{T}}$ satisfies $(\varepsilon, \delta)$-SDP.

**Regret.** By the same analysis in the proof of Theorem 3.5 in Chowdhury & Zhou (2022b), the injected noise for each calculation of the noisy *synchronized* p-sum is sub-Gaussian with the variance being at most $\widehat{\sigma}^2 = O\left( \frac{\log^2(d^2/\delta_0)}{\varepsilon_0^2} \right) = O\left( \frac{\kappa \log(1/\delta) \log^2(d^2\kappa/\delta)}{\varepsilon^2} \right)$. Now, by the binary tree structure, each prefix sum only involves at most $\kappa$ p-sums. Hence, the overall noise level is upper bounded by $\sigma_{\text{total}}^2 = \kappa\widehat{\sigma}^2$. Finally, setting $\sigma^2$ in Lemma C.4 to be the noise level $\sigma_{\text{total}}^2$, yields the required result. □

Now, we provide proof of amplification Lemma F.2 for completeness. We follow the same idea as in Feldman et al. (2022) and Lowy & Razaviyayn (2021). For easy comparison, we use the same notations as in Lowy & Razaviyayn (2021) and highlighted the key difference using color text.

*Proof of Lemma F.2.* Let $\mathbf{X}_0, \mathbf{X}_1 \in \mathcal{X}^{n \times N}$ be adjacent distributed data sets (i.e. $\sum_{i=1}^{N} \sum_{j=1}^{n} \mathbb{1}_{\{x_{i,j} \neq x_{i,j}\}} = 1$). Assume WLOG that $\mathbf{X}_0 = (X_1^0, X_2, \cdots, X_N)$ and $\mathbf{X}_1 = (X_1^1, X_2, \cdots, X_N)$, where $X_1^0 = (x_{1,0}, x_{1,2}, \cdots, x_{1,n}) \neq (x_{1,1}, x_{1,2}, \cdots, x_{1,n})$. We can also assume WLOG that $X_j \notin \{X_1^0, X_1^1\}$ for all $j \in \{2, \cdots, N\}$ by re-defining $\mathcal{X}$ and $\mathcal{R}_r^{(i)}$ if necessary.

Fix $i \in [N], r \in [R], \mathbf{Z} = \mathbf{Z}_{1:r-1} = Z_{(1:r-1)}^{(1:N)} \in \mathcal{Z}^{(r-1) \times N}$, denote $\mathcal{R}(X) := \mathcal{R}_r^{(i)}(\mathbf{Z}, X)$ for $X \in \mathcal{X}^n$, and $\mathcal{A}_s(\mathbf{X}) := \mathcal{A}_s^r(\mathbf{Z}_{1:r-1}, \mathbf{X})$. Draw $\pi$ uniformly from the set of permutations of $[N]$. Now, since $\mathcal{R}$ is $(\varepsilon_0^r, \delta_0^r)$-DP, $\mathcal{R}(X_1^1) \underset{(\varepsilon_0^r, \delta_0^r)}{\simeq} \mathcal{R}(X_1^0)$, so by Lowy & Razaviyayn (2021, Lemma D.12), there exists a local randomizer $\mathcal{R}'$ such that $\mathcal{R}'(X_1^1) \underset{(\varepsilon_0^r, 0)}{\simeq} \mathcal{R}(X_1^0)$ and $TV(\mathcal{R}'(X_1^1), \mathcal{R}(X_1^1)) \leqslant \delta_0^r$.

Hence, by Lowy & Razaviyayn (2021, Lemma D.8), there exist distributions $U(X_1^0)$ and $U(X_1^1)$ such that

$$\mathcal{R}(X_1^0) = \frac{e^{\varepsilon_0^r}}{e^{\varepsilon_0^r} + 1} U(X_1^0) + \frac{1}{e^{\varepsilon_0^r} + 1} U(X_1^1) \tag{11}$$

and

$$\mathcal{R}'(X_1^1) = \frac{1}{e^{\varepsilon_0^r} + 1} U(X_1^0) + \frac{e^{\varepsilon_0^r}}{e^{\varepsilon_0^r} + 1} U(X_1^1). \tag{12}$$

Here, we diverge from the proof in (Lowy & Razaviyayn, 2021). We denote $\widetilde{\varepsilon}_0 := n\varepsilon_0^r$ and $\widetilde{\delta}_0 := \delta_0^r$. Then, by the assumption of $\mathcal{R}(X)$, for any $X$, we have $\mathcal{R}(X) \underset{(\widetilde{\varepsilon}_0, \widetilde{\delta}_0)}{\simeq} \mathcal{R}(X_1^0))$ and $\mathcal{R}(X) \underset{(\widetilde{\varepsilon}_0, \widetilde{\delta}_0)}{\simeq} \mathcal{R}(X_1^1))$. This is because by the assumption, when the dataset changes from any $X$ to $X_1^0$ (or $X_1^1$), the total change in terms of $l_2$ norm can be $n$ times that under an adjacent pair. Thus, one has to scale the $\varepsilon_0^r$ by $n$ while keeping the same $\delta_0^r$.

Now, we resume the same idea as in (Lowy & Razaviyayn, 2021). By convexity of hockey-stick divergence and the above result, we have $\mathcal{R}(X) \underset{(\widetilde{\varepsilon}_0, \widetilde{\delta}_0)}{\simeq} \frac{1}{2}(\mathcal{R}(X_1^0) + \mathcal{R}(X_1^1)) := \rho$ for all $X \in \mathcal{X}^n$.

That is, $\mathcal{R}$ is $(\widetilde{\varepsilon}_0, \widetilde{\delta}_0)$ deletion group DP for groups of size $n$ with reference distribution $\rho$. Thus, by Lowy & Razaviyayn (2021, Lemma D.11), we have that there exists a local randomizer $\mathcal{R}''$ such that $\mathcal{R}''(X)$ and $\rho$ are $(\widetilde{\varepsilon}_0, 0)$ indistinguishable and $TV(\mathcal{R}''(X), \mathcal{R}(X)) \leqslant \widetilde{\delta}_0$ for all $X$. Then by the definition of $(\widetilde{\varepsilon}_0, 0)$ indistinguishability, for all $X$ there exists a "left-over" distribution $LO(X)$ such that $\mathcal{R}''(X) = \frac{1}{e^{\widetilde{\varepsilon}_0}}\rho + (1 - 1/e^{\widetilde{\varepsilon}_0})LO(X) = \frac{1}{2e^{\widetilde{\varepsilon}_0}}(\mathcal{R}(X_1^0) + \mathcal{R}(X_1^1)) + (1 - 1/e^{\widetilde{\varepsilon}_0})LO(X)$.

Now, define a randomizer $\mathcal{L}$ by $\mathcal{L}(X_1^0) := \mathcal{R}(X_1^0)$, $\mathcal{L}(X_1^1) := \mathcal{R}'(X_1^1)$, and

$$\mathcal{L}(X) := \frac{1}{2e^{\widetilde{\varepsilon}_0}}\mathcal{R}(X_1^0) + \frac{1}{2e^{\widetilde{\varepsilon}_0}}\mathcal{R}'(X_1^1) + (1 - 1/e^{\widetilde{\varepsilon}_0})LO(X)$$

$$= \frac{1}{2e^{\widetilde{\varepsilon}_0}}U(X_1^0) + \frac{1}{2e^{\widetilde{\varepsilon}_0}}U(X_1^1) + (1 - 1/e^{\widetilde{\varepsilon}_0})LO(X) \tag{13}$$

for all $X \in \mathcal{X}^n \setminus \{X_1^0, X_1^1\}$. (The equality follows from (11) and (12).) Note that $TV(\mathcal{R}(X_1^0), \mathcal{L}(X_1^0)) = 0$, $TV(\mathcal{R}(X_1^1), \mathcal{L}(X_1^1)) \leqslant \delta_0^r$, and for all $X \in \mathcal{X}^n \setminus \{X_1^0, X_1^1\}$, $TV(\mathcal{R}(X), \mathcal{L}(X)) \leqslant TV(\mathcal{R}(X), \mathcal{R}''(X)) + TV(\mathcal{R}''(X), \mathcal{L}(X)) \leqslant \widetilde{\delta}_0 + \frac{1}{2e^{\widetilde{\varepsilon}_0}}TV(\mathcal{R}'(X_1^1), \mathcal{R}(X_1^1)) = (1 + \frac{1}{2e^{n\varepsilon_0^r}})\delta_0^r$.

Keeping $r$ fixed (omitting $r$ scripts everywhere), for any $i \in [N]$ and $\mathbf{Z} := \mathbf{Z}_{1:r-1} \in \mathcal{Z}^{(r-1) \times N}$, let $\mathcal{L}^{(i)}(\mathbf{Z}, \cdot), U^{(i)}(\mathbf{Z}, \cdot)$, and $LO^{(i)}(\mathbf{Z}, \cdot)$ denote the randomizers resulting from the process described above. Let $\mathcal{A}_{\mathcal{L}} : \mathcal{X}^{n \times N} \to \mathcal{Z}^N$ be defined exactly the same way as $\mathcal{A}_s^r := \mathcal{A}_s$ (same $\pi$) but with the randomizers $\mathcal{R}^{(i)}$ replaced by $\mathcal{L}^{(i)}$. Since $\mathcal{A}_s$ applies each randomizer $\mathcal{R}^{(i)}$ exactly once and $\mathcal{R}^{(1)}(\mathbf{Z}, X_{\pi(1)}), \cdots \mathcal{R}^{(N)}(\mathbf{Z}, X_{\pi(N)})$ are independent (conditional on $\mathbf{Z} = \mathbf{Z}_{1:r-1}$) [10], we have $TV(\mathcal{A}_s(\mathbf{X}_0), \mathcal{A}_{\mathcal{L}}(\mathbf{X}_0)) \leqslant N(1 + \frac{1}{2e^{n\varepsilon_0^r}})\delta_0^r$ and $TV(\mathcal{A}_s(\mathbf{X}_1), \mathcal{A}_{\mathcal{L}}(\mathbf{X}_1)) \leqslant N(1 + \frac{1}{2e^{n\varepsilon_0^r}})\delta_0^r$. Now we

---

[10]This follows from the assumption that $\mathcal{R}^{(i)}(\mathbf{Z}_{1:r-1}, X)$ is conditionally independent of $X'$ given $\mathbf{Z}_{1:r-1}$ for all $\mathbf{Z}_{1:r-1}$ and $X \neq X'$.

claim that $\mathcal{A}_{\mathcal{L}}(\mathbf{X}_0)$ and $\mathcal{A}_{\mathcal{L}}(\mathbf{X}_1)$ are $(\varepsilon^r, \delta)$ indistinguishable for any $\delta \geqslant 2e^{-Ne^{-n\varepsilon_0^r}/16}$. Observe that this claim implies that $\mathcal{A}_s(\mathbf{X}_0)$ and $\mathcal{A}_s(\mathbf{X}_1)$ are $(\varepsilon^r, \delta^r)$ indistinguishable by Lowy & Razaviyayn (2021, Lemma D.13) (with $P' := \mathcal{A}_{\mathcal{L}}(\mathbf{X}_0), Q' := \mathcal{A}_{\mathcal{L}}(\mathbf{X}_1), P := \mathcal{A}_s(\mathbf{X}_0), Q := \mathcal{A}_s(\mathbf{X}_1)$.) Therefore, it only remains to prove the claim, i.e. to show that $D_{e^{\varepsilon^r}}(\mathcal{A}_{\mathcal{L}}(\mathbf{X}_0), \mathcal{A}_{\mathcal{L}}(\mathbf{X}_1) \leqslant \delta$ for any $\delta \geqslant 2e^{-Ne^{-n\varepsilon_0^r}/16}$.

Now, define $\mathcal{L}_U^{(i)}(\mathbf{Z}, X) := \begin{cases} U^{(i)}(\mathbf{Z}, X_1^0) & \text{if } X = X_1^0 \\ U^{(i)}(\mathbf{Z}, X_1^1) & \text{if } X = X_1^1 \\ \mathcal{L}^{(i)}(\mathbf{Z}, X) & \text{otherwise.} \end{cases}$ For any inputs $\mathbf{Z}, \mathbf{X}$, let $\mathcal{A}_U(\mathbf{Z}, \mathbf{X})$ be

defined exactly the same as $\mathcal{A}_s(\mathbf{Z}, \mathbf{X})$ (same $\pi$) but with the randomizers $\mathcal{R}^{(i)}$ replaced by $\mathcal{L}_U^{(i)}$. Then by (11) and (12),

$$\mathcal{A}_{\mathcal{L}}(\mathbf{X}_0) = \frac{e^{\varepsilon_0^r}}{e^{\varepsilon_0^r}+1}\mathcal{A}_U(\mathbf{X}_0) + \frac{1}{e^{\varepsilon_0^r}+1}\mathcal{A}_U(\mathbf{X}_1) \text{ and } \mathcal{A}_{\mathcal{L}}(\mathbf{X}_1) = \frac{1}{e^{\varepsilon_0^r}+1}\mathcal{A}_U(\mathbf{X}_0) + \frac{e^{\varepsilon_0^r}}{e^{\varepsilon_0^r}+1}\mathcal{A}_U(\mathbf{X}_1).$$
(14)

Then by (13), for any $X \in \mathcal{X}^n \setminus \{X_1^0, X_1^1\}$ and any $\mathbf{Z} = \mathbf{Z}_{1:r-1} \in \mathcal{Z}^{(r-1)\times N}$, we have $\mathcal{L}_U^{(i)}(\mathbf{Z}, X) = \frac{1}{2e^{\widetilde{\varepsilon}_0}}\mathcal{L}_U^{(i)}(\mathbf{Z}, X_1^0) + \frac{1}{2e^{\widetilde{\varepsilon}_0}}\mathcal{L}_U^{(i)}(\mathbf{Z}, X_1^1) + (1 - e^{-\widetilde{\varepsilon}_0})LO^{(i)}(\mathbf{Z}, X)$. Hence, Lowy & Razaviyayn (2021, Lemma D.10) (with $p := e^{-\widetilde{\varepsilon}_0} = e^{-n\varepsilon_0^r}$) implies that $\mathcal{A}_U(\mathbf{X}_0)$ and $\mathcal{A}_U(\mathbf{X}_1)$) are

$$\left( \log\left(1 + \frac{8\sqrt{e^{\widetilde{\varepsilon}_0}\ln(4/\delta)}}{\sqrt{N}} + \frac{8e^{\widetilde{\varepsilon}_0}}{N}\right), \delta \right)$$

indistinguishable for any $\delta \geqslant 2e^{-Ne^{-n\varepsilon_0^r}/16}$.

Here, we also slightly diverge from Lowy & Razaviyayn (2021). Instead of using Lowy & Razaviyayn (2021, Lemma D.14), we can directly follow the proof of Lemma 3.5 in Feldman et al. (2022) and Lemma 2.3 in Feldman et al. (2022) to establish our claim that $\mathcal{A}_{\mathcal{L}}(\mathbf{X}_0)$ and $\mathcal{A}_{\mathcal{L}}(\mathbf{X}_1)$ are indistinguishable (hence the final result). Here, we also slightly improve the $\delta$ term compared to Feldman et al. (2022) by applying amplification via sub-sampling to the $\delta$ term as well. In particular, the key step is to rewrite (14) as follows (with $T := \frac{1}{2}(\mathcal{A}_U(\mathbf{X}_0) + \mathcal{A}_U(\mathbf{X}_1))$

$$\mathcal{A}_{\mathcal{L}}(\mathbf{X}_0) = \frac{2}{e^{\varepsilon_0^r}+1}T + \frac{e^{\varepsilon_0^r}-1}{e^{\varepsilon_0^r}+1}\mathcal{A}_U(\mathbf{X}_0) \text{ and } \mathcal{A}_{\mathcal{L}}(\mathbf{X}_1) = \frac{2}{e^{\varepsilon_0^r}+1}T + \frac{e^{\varepsilon_0^r}-1}{e^{\varepsilon_0^r}+1}\mathcal{A}_U(\mathbf{X}_1). \quad (15)$$

Thus, by the convexity of the hockey-stick divergence and Lemma 2.3 in Feldman et al. (2022), we have $\mathcal{A}_{\mathcal{L}}(\mathbf{X}_0)$ and $\mathcal{A}_{\mathcal{L}}(\mathbf{X}_1)$ are

$$\left( \log\left(1 + \frac{\varepsilon_0^r-1}{\varepsilon_0^r+1}\left(\frac{8\sqrt{e^{\widetilde{\varepsilon}_0}\ln(4/\delta^r)}}{\sqrt{N}}\right) + \frac{8e^{\widetilde{\varepsilon}_0}}{N}\right), \frac{\varepsilon_0^r-1}{\varepsilon_0^r+1}\delta \right)$$

indistinguishable for any $\delta \geqslant 2e^{-Ne^{-n\varepsilon_0^r}/16}$. As decribed before, this leads to the result that $\mathcal{A}_s(\mathbf{X}_0)$ and $\mathcal{A}_s(\mathbf{X}_1)$ are $(\varepsilon^r, \delta^r)$ indistinguishable by Lowy & Razaviyayn (2021, Lemma D.13) (original result in Lemma 3.17 of Dwork & Roth (2014)) with (noting that $\widetilde{\varepsilon}_0 = n\varepsilon_0^r$)

$$\varepsilon^r := \ln\left[1 + \left(\frac{e^{\varepsilon_0^r}-1}{e^{\varepsilon_0^r}+1}\right)\left(\frac{8\sqrt{e^{n\varepsilon_0^r}\ln(4/\delta)}}{\sqrt{N}} + \frac{8e^{n\varepsilon_0^r}}{N}\right)\right],$$

$$\delta^r := \left(\frac{e^{\varepsilon_0^r}-1}{e^{\varepsilon_0^r}+1}\right)\delta + N(e^{\varepsilon^r}+1)(1 + e^{-\varepsilon_0^r}/2)\delta_0^r.$$

$\square$

# G  FURTHER DISCUSSIONS

In this section, we provide more details on our upper bounds, privacy notion and algorithm design.

## G.1  DISCUSSION ON TIGHTNESS OF UPPER BOUNDS

In the paper, we have established regrets of $O(M^{3/4}\sqrt{T/\varepsilon})$ under silo-level LDP and $O(\sqrt{MT/\varepsilon})$ under SDP. An essential open question is regarding tightness of these upper bounds. It turns out that

the key to obtain both lower bounds is to first establish a tight characterization for single-agent LCB under central JDP (which is still open to the best of our knowledge), as elaborated below:

**SDP:** The current upper bound aligns with the state-of-the-art result achieved by a super single agent under central JDP. However, the tightness of this bound is still uncertain, as it even remains open whether the upper bound under the centralized setting is tight. To our best knowledge, the only existing lower bound for LCBs under central JDP is $\Omega(\sqrt{T} + 1/\varepsilon)$ (He et al., 2022b), implying a lower bound of $\Omega(\sqrt{MT} + 1/\varepsilon)$ for the super single agent case. That is, the privacy cost in the lower bound is only additive rather than multiplicative cost of $1/\sqrt{\varepsilon}$ present in the upper bound. It is unclear to us which one of the upper bound or lower bound is loose.

**Silo-level LDP:** A lower bound can potentially be established using a similar reduction as in Lowy & Razaviyayn (2021), where the authors derive a lower bound for silo-level LDP supervised learning via the central model. To be more specific, for any silo-level LDP algorithm $\mathcal{A}$ with privacy guarantee $\varepsilon_0$, one can first "virtually" shuffle all the $MT$ user sequences and then apply $\mathcal{A}$, leading to a shuffled version $\mathcal{A}_s$. As shown in Lowy & Razaviyayn (2021),, the shuffled version algorithm $\mathcal{A}_s$ enjoys an SDP privacy guarantee of roughly $\varepsilon := \varepsilon_0/\sqrt{M}$ (here again, one cannot directly use standard amplification lemma). Since SDP implies central JDP in our linear contextual bandit case, then one can conclude that $\mathcal{A}_s$ has a lower bound of $L_c(\varepsilon)$, where $L_c(\varepsilon)$ denotes the lower bound for LCB under central JDP with privacy guarantee $\varepsilon$. Finally, one can note that $\mathcal{A}$ and $\mathcal{A}_s$ have the same regret performance, hence establishing the regret lower bound $L_c(\varepsilon_0/\sqrt{M})$ for $\mathcal{A}$ under silo-level LDP.

**Implication:** If one can establish a lower bound $L_c(\varepsilon) = \Omega(\sqrt{T/\varepsilon})$ for standard LCB under central JDP (i.e., $\Omega(\sqrt{MT/\varepsilon})$ for the super single agent case), then by the above argument, it directly implies that our SDP upper bound is tight and moreover, the upper bound under silo-level LDP is also tight.

It is worth noting that our new findings in this paper (e.g., identifying existing gaps and establishing new upper bounds) motivate the above interesting questions, which we believe will promote advances in the field.

### G.2 SILO-LEVEL LDP/SDP VS. OTHER PRIVACY NOTIONS

In this section, we compare our silo-level LDP and SDP with standard privacy notions for single-agent LCBs, including local, central, and shuffle model for DP, respectively.

**Silo-level LDP vs. single-agent local DP.** Under standard LDP for single-agent LCBs (Zheng et al., 2020; Duchi et al., 2013; Zhou & Tan, 2021), each user only trusts herself and hence privatizes her response before sending it to the agent. In contrast, under silo-level LDP, each local user trusts the local silo (agent), which aligns with the pratical situations of cross-silo FL, e.g., patients often trust the local hospitals. In such cases, standard LDP becomes unnecessarily stringent, hindering performance/regret and making it less appealing to cross-silo federated LCBs.

**Silo-level LDP vs. single-agent central DP.** The comparison with standard central DP (in particular central JDP)[11] for single-agent LCB (e.g., Shariff & Sheffet (2018)) is delicate. We first note that under both notions, users trust the agent and the privacy burden lies at the agent. Under standard central DP, the agent uses private statistics until round $t$ to choose action for each round $t$, which ensures that any other users $t' \neq t$ cannot infer too much about user $t$'s information by observing the actions on rounds $t' \neq t$ (i.e., joint differential privacy (JDP) (Kearns et al., 2014)). On the other hand, silo-level LDP does not necessarily require each agent (silo) to use private statistics to recommend actions to users within the silo. Instead, it only requires the agent to privatize its sent messages (both schedule and content). Thus, silo-level LDP may not protect a user $t$ from the colluding of all other users within the *same* silo. In other words, the adversary model for silo-level LDP is that the adversary could be any other silos or the central server rather than other users within the same silo. Note that the same adversary model is assumed in a similar notion for federated supervised learning (e.g., inter-silo record-level differential privacy (ISRL-DP) in Lowy & Razaviyayn (2021)). In fact, with a minor tweak of our Algorithm 1, one can achieve a slightly stronger notion of privacy than silo-level LDP in that it now can protect against both other silos/server and users within the same silo.

---

[11]As shown in Shariff & Sheffet (2018), JDP relaxation is necessary for achieving sub-linear regret for LCBs under the central model. Otherwise, a linear regret lower bound exists for central standard DP.

The key idea is exactly that now each agent will only use private statistics to recommend actions, see Appendix G.3.

**Silo-level LDP vs. Federated DP in Dubey & Pentland (2020).** In Dubey & Pentland (2020), the authors define the so-called notion of *federated DP* for federated LCBs, which essentially means that "the action chosen by any agent must be sufficiently impervious (in probability) to any single data from any other agent". This privacy guarantee is directly implied by our silo-level LDP. In fact, in order to show such a privacy guarantee, Dubey & Pentland (2020) basically tried to show that the outgoing communication is private, which is the idea of silo-level LDP. However, as mentioned in the main paper, Dubey & Pentland (2020) only privatizes the communicated data and fails to privatize the communication schedule, which leads to privacy leakage. Moreover, as already mentioned in Remark B.1, Fed-DP fails to protect a user's privacy even under a reasonable adversary model. Thus, we believe that silo-level LDP is a better option for federated LCBs.

**SDP vs. single-agent shuffle DP.** Under the single-agent shuffle DP (Chowdhury & Zhou, 2022b; Tenenbaum et al., 2023), the shuffler takes as input a batch of users' data (i.e., from $t_1$ to $t_2$), which enables to achieve a regret of $\widetilde{O}(T^{3/5})$ (vs. $\widetilde{O}(T^{3/4})$ regret under local model and $\widetilde{O}(\sqrt{T})$ regret under central model). In contrast, under our SDP, the shuffler takes as input the DP outputs from all $M$ agents. Roughly speaking, single-agent shuffle DP aims to amplify the privacy dependence on $T$ while our SDP amplifies privacy over $M$. Due to this, single-agent shuffle DP can directly apply a standard amplification lemma (e.g., Feldman et al. (2022)) or shuffle protocol (e.g., Cheu et al. (2021)) that works well with LDP mechanism at each user (i.e., the size of dataset is $n = 1$). In contrast, in order to realize amplification over $M$ agents' DP outputs, we have to carefully modify the standard amplification lemma to handle the fact that now each local mechanism operates on $n > 1$ data points, which is one of the key motivations for our new amplification lemma.

**Sublinear regret under SDP vs. linear regret lower bound under central standard DP (not JDP).** One may wonder why an even better regret under the shuffle model is possible given that there is a linear regret bound under central model for LCBs. This is not contradicting as the lower bound of linear regret is established under standard central DP while SDP in LCBs only implies central JDP. More specifically, in contrast to standard private data analysis and supervised learning, the shuffle model is NOT an intermediate trust model between central standard DP and local DP for LCBs. That is, even if an LCB algorithm satisfies shuffle DP, it can still fail to satisfy central standard DP. Rather, it is only an intermediate trust model between the central joint DP (JDP) and the local model. That is, if an LCB algorithm satisfies shuffle DP, it satisfies central JDP (via Billboard lemma).

### G.3    A SIMPLE TWEAK OF ALGORITHM 1 FOR A STRONGER PRIVACY GUARANTEE

As discussed in the last subsection, the adversary model behind silo-level LDP only includes other silos and the central server, i.e., excluding adversary users within the same silo. Thus, for silo-level LDP, Algorithm 1 can use non-private data to recommend actions within a batch (e.g., $V_{t,i}$ includes non-private recent local bias vectors and covariance matrices). If one is also interested in protecting against adversary users within the same silo, a simple tweak of Algorithm 1 suffices.

As shown in Algorithm 8, the only difference is a lazy update of $\widehat{\theta}_{t,i}$ is adopted (line 5), i.e., it is only computed using private data without any dependence on new non-private local data. In fact, same regret bound as in Theorem 5.1 can be achieved for this new algorithm (though empirical performance could be worse due to the lazy update). In the following, we highlight the key changes in the regret analysis. It basically follows the six steps in the proof of Lemma C.2. One can now define a mapping $\kappa(t)$ that maps any $t \in [T]$ to the most recent communication round. That is, for any $t \in [t_{k-1}, t_k]$ where $t_k = kB$ is the $k$-th communication round, we have $\kappa(t) = t_{k-1}$. Then, one can replace all $t$ in $V_{t,i}$ and $G_{t,i}$ by $\kappa(t)$. The main difference that needs a check is Step 4 when bounding the regret in good epochs. The key is again to establish a similar form as (5). To this end, note that for all $t \in [t_{k-1}, t_k]$ $V_k \succeq \bar{V}_{t,i}$ and $G_{\kappa(t),i} + \lambda_{\min} I = V_{k-1}$, which enables us to obtain $\|x_{t,i}\|_{(G_{\kappa(t),i} + \lambda_{\min} I)^{-1}} \leq \sqrt{2} \|x_{t,i}\|_{\bar{V}_{t,i}^{-1}}$. Following the same analysis yields the desired regret bound.

---

**Algorithm 8** Priv-FedLinUCB-Lazy

---

1: **Parameters:** Batch size $B \in \mathbb{N}$, regularization $\lambda > 0$, confidence radii $\{\beta_{t,i}\}_{t\in[T],i\in[M]}$, feature map $\phi_i : \mathcal{C}_i \times \mathcal{K}_i \to \mathbb{R}^d$, privacy protocol $\mathcal{P} = (\mathcal{R}, \mathcal{S}, \mathcal{A})$

2: **Initialize:** For all $i \in [M]$, $W_i = 0, U_i = 0, \widetilde{W}_{\text{syn}} = 0, \widetilde{U}_{\text{syn}} = 0$

3: **for** $t = 1, \ldots, T$ **do**

4:     **for** each agent $i = 1, \ldots, M$ **do**

5:         $V_{t,i} = \lambda I + \widetilde{W}_{\text{syn}}, \widehat{\theta}_{t,i} = V_{t,i}^{-1}\widetilde{U}_{\text{syn}}$

6:         Play arm $a_{t,i} = \text{argmax}_{a\in\mathcal{K}_i}\langle\phi_i(c_{t,i},a),\widehat{\theta}_{t,i}\rangle + \beta_{t,i}\|\phi_i(c_{t,i},a)\|_{V_{t,i}^{-1}}$ and set $x_{t,i} = \phi_i(c_{t,i},a_{t,i})$

7:         Observe reward $y_{t,i}$

8:         Update $U_i = U_i + x_{t,i}y_{t,i}$ and $W_i = W_i + x_{t,i}x_{t,i}^\top$

9:     **end for**

10:    **if** $t \bmod B = 0$ **then**

11:       // Local randomizer $\mathcal{R}$ at *all* agents $i \in [M]$

12:       Send randomized messages $R_{t,i}^{\text{bias}} = \mathcal{R}^{\text{bias}}(U_i)$ and $R_{t,j}^{\text{cov}} = \mathcal{R}^{\text{cov}}(W_i)$ to the shuffler

13:       // Third party $\mathcal{S}$

14:       $S_t^{\text{bias}} = \mathcal{S}(\{R_{t,i}^{\text{bias}}\}_{i\in[M]})$ and $S_t^{\text{cov}} = \mathcal{S}(\{R_{t,i}^{\text{cov}}\}_{i\in[M]})$

15:       // Analyzer $\mathcal{A}$ at the server

16:       Construct private cumulative statistics $\widetilde{U}_{\text{syn}} = \mathcal{A}^{\text{bias}}(S_t^{\text{bias}})$ and $\widetilde{W}_{\text{syn}} = \mathcal{A}^{\text{cov}}(S_t^{\text{cov}})$

17:       // *All* agents $i \in [M]$

18:       Receive $\widetilde{W}_{\text{syn}}$ and $\widetilde{U}_{\text{syn}}$ from the server

19:       Reset $W_i = 0, U_i = 0$

20:    **end if**

21: **end for**

---

### G.4 NON-UNIQUE USERS

In the main paper, we assume all users across all silos and $T$ rounds are unique. Here, we briefly discuss how to handle the case of non-unique users.

- The same user appears multiple times in the same silo. One example of this could be one patient visiting the same hospital multiple times. In such cases, one needs to carefully apply group privacy or other technique (e.g., Chowdhury & Zhou (2022b)) to characterize the privacy loss of these returning users.

- The same user appears multiple times across different silos. One example of this could be one patient who has multiple records across different hospitals. Then, one needs to use adaptive advanced composition to characterize the privacy loss of these returning users.

## H ADDITIONAL DETAILS ON SIMULATION RESULTS

In Figure 3, we compare regret performance of LDP-FedLinUCB with FedLinUCB under varying privacy budgets.[12] In sub-figure (a), we plot results for $\delta = 0.1$ and varying level of $\varepsilon \in \{0.2, 1, 5\}$ on synthetic Gaussian bandit instance, wherein sub-figure (b), we plot results for $\varepsilon = 5$ and varying level of $\delta \in \{0.1, 0.01, 0.001\}$. In sub-figure (c), we plot results for $\delta = 0.1$ and varying level of $\varepsilon \in \{0.2, 1, 5\}$ on bandit instance generated from MSLR-WEB10K data by training a lasso model on bodyfeatures ($d = 78$). In all these plots, we observe that regret of LDP-FedLinUCB decreases and, comes closer to that of FedLinUCB as $\varepsilon, \delta$ increases (i.e., level of privacy protection decreases), which support our theoretical results. Here, we don't compare SDP-LinUCB (with privacy amplification) since its privacy guarantee holds for $\varepsilon, \delta \ll 1$. Instead, we do so in sub-figure (d) with $\varepsilon = \delta = 0.0001$. Here also, we observe a drop in regret of SDP-FedLinUCB compared to that of LDP-FedLinUCB.

---

[12] All existing non-private federated LCB algorithms (e.g., Wang et al. (2020)) adopts adaptive communication. We refrain from comparing with those to maintain consistency in presentation.

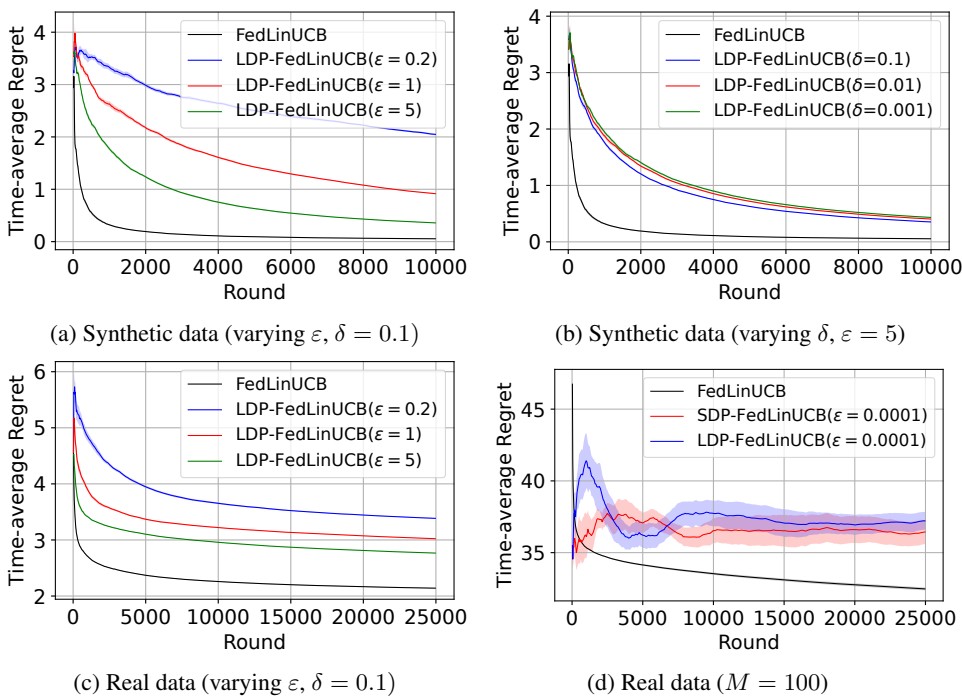

(a) Synthetic data (varying $\varepsilon$, $\delta = 0.1$)

(b) Synthetic data (varying $\delta$, $\varepsilon = 5$)

(c) Real data (varying $\varepsilon$, $\delta = 0.1$)

(d) Real data ($M = 100$)

Figure 3: Comparison of time-average group regret for FedLinUCB (non-private) and LDP-FedLinUCB (i.e., under silo-level LDP) on (a, b) synthetic Gaussian bandit instance and (c,d) bandit instance generated from MSLR-WEB10K Learning to Rank dataset.

