# OpenReview forum: "On Differentially Private Federated Linear Contextual Bandits"
_ICLR.cc/2024/Conference — ICLR 2024 poster_

### Official Review · Reviewer_cqha · 2023-10-28

**Soundness:** 2 fair
**Presentation:** 4 excellent
**Contribution:** 3 good
**Rating:** 6
**Confidence:** 4

**Summary:**

This paper delves into the exploration of federated learning in the context of linear contextual bandits (LCBs) while incorporating the principles of differential privacy (DP). The proposed algorithmic framework encompasses several key components, including LinUCB exploration, a distributed variant of the tree-based mechanism, privacy amplification, and a fixed batch update approach. This comprehensive approach offers compelling solutions for addressing both silo-level local DP and shuffle DP concerns. Additionally, the authors have identified and rectified an error pertaining to the total injected privacy noise, as previously reported in Dubey & Pentland 2020, contributing to the advancement of this field.

**Strengths:**

1. The analysis in this paper is notably meticulous, particularly in its identification of the mistake in the previous results. The presentation of the findings is commendable, as it effectively illustrates the potential privacy vulnerabilities present in Dubey & Pentland 2020. Furthermore, it compellingly highlights the necessity of a more robust silo-level local DP setting. The comparison with related work is both thorough and detailed, contributing to a well-rounded understanding of the research landscape.

2. The paper provides a comprehensive narrative that encompasses regret and communication guarantees under varying privacy constraints. Impressively, the authors extend these guarantees to a broader range of privacy parameters, demonstrating a thorough exploration of the subject matter. Moreover, the novel extension of the current amplification analysis to their specific case adds a valuable dimension to the research.

**Weaknesses:**

1. The paper, while containing valuable insights, may benefit from some improvements in terms of clarity and presentation. Firstly, the algorithmic design, which incorporates several components and explores various privacy settings, could be made more accessible by summarizing the results in a table format. This would provide readers with a quick and clear overview of corresponding regret, communication, and algorithmic elements across different settings. Additionally, in terms of organization, it might be helpful to reconsider the placement of the concluding remarks, which currently reside under the simulation results. Furthermore, there appears to be a spacing issue in the section discussing the tree-based mechanism, possibly resulting from the authors' use of the vspace command in LaTeX.

2. While the paper offers valuable contributions, the novelty and inspiration of the problem itself could be further developed. The concept of silo-level Local Differential Privacy (LDP), while important and rigorous, may be considered relatively straightforward to formulate. Additionally, the solutions presented in the paper largely draw upon existing results and algorithmic designs, rather than introducing fundamentally new insights or modifications to established findings, which limits the potential of this paper.

**Questions:**

None.

---

> ### Author Response · Authors · 2023-11-13
> **Response**
>
> We thank the reviewer for reviewing our paper and providing great suggestions on our writing.
>
> **Summary table of main results:** Thanks for the great suggestion. We have added a summary table in the introduction to summarize our main results.
>
> **Placement of the concluding remarks and use of vspace:** Fixed both. The conclusion is now a separate section in our new version.
>
> **On our new results:** We agree with the reviewer that silo-level LDP and SDP are relatively straightforward to formulate since these notions are widely adopted in supervised learning problems. In this work, we adapt them to a sequential learning problem of linear contextual bandits.
>
> However, in order to establsih our regret bounds, we have introduced two new results, since directly applying exsiting techniques or algorithms for supervised learning do not work in sequential learning.
>
>
> - First, we point out that simply adding an additional shuffler between the agents and server to Algorithm 1 in Dubey & Pentland, 2020  is not sufficient to achieve "optimal" regret under SDP. Instead, the shuffle mechanism must be thoughtfully integrated into the tree-based algorithm to shuffle the p-sums rather than prefix sums (which is the case for standard shuffle mechanism).
>
> - Second, we highlight that one cannot merely rely on existing privacy amplification lemmas to achieve SDP. We have derived a **new result** on privacy amplification for shuffling DP mechanisms where the dataset size is greater than one, as opposed to the standard results where the dataset size is equal to one.
>
> Finally, for the regret analysis, our approach to bound the "local drift" under fixed-batch schedule is different from the approach under adaptive schedule.

---

> > ### Comment · Reviewer_cqha · 2023-11-22
> >
> > The answers address the reviewer's concerns and the reviewer would like to keep the positive scores for this work.

---

### Official Review · Reviewer_Gszt · 2023-10-31

**Soundness:** 3 good
**Presentation:** 3 good
**Contribution:** 3 good
**Rating:** 8
**Confidence:** 3

**Summary:**

This paper Investigates linear contextual bandits user the cross-silo DP. It seems that this paper is motivated by clear holes in the work of Dubey and Petland (2020), which is a highly cited paper. This work succinctly identifies the errors in that paper, as well as proposes their own solution. It is quite remarkable, since that work is well cited, but the arguments of the authors seem convincing to me.

In addition to showing the errors with that work, this work further develops a variant of LinUCB that provides the required level of privacy, and computes the regret bound. Another variant is considered where shuffle DP is used, which enables regret equal to the regret achieved by the super node.

**Strengths:**

* Important to set the record straight if errors in Dubey and Petland (2020) is not well known
* Intuitive algorithms that build on well studied baselines
* Generally good flow and writing
* Thorough treatment, including shuffle DP setting as well

**Weaknesses:**

I think related work sections should be in the main body of the paper for a 9 page paper

**Questions:**

Since there is so much discussion of Dubey and Petland (2020), I would like to see some if the information related to it included in the abstract.

I find Section 3 to be a bit prose heavy.

---

> ### Author Response · Authors · 2023-11-13
> **Response**
>
> Thanks for your positive evaluation of our paper and great suggestions on writing!
>
> **Put related work in the main paper:** Thanks for the suggestion. We have included related work in the main paper (highlighted in brown color on page 2). We also provide further discussions on related work in Appendix A.
>
> **On Dubey and Petland (2020) in the abstract:** Thanks for the suggestion. We have already pointed out all three issues pertaining to Dubey and Petland (2020)  in the abstract. In our new version, we have now added a reference to Dubey and Petland (2020) in the abstract.
>
> **On the writing of Section 3:** We have carefully rewritten Section 3 to improve its readability in our new version.

---

### Official Review · Reviewer_kYUa · 2023-10-31

**Soundness:** 4 excellent
**Presentation:** 4 excellent
**Contribution:** 3 good
**Rating:** 8
**Confidence:** 3

**Summary:**

This paper considers silo-level LDP and SDP in federated contextual bandit problems. It points out the existing gap in previous LDP federated bandit works and provides new approaches with regret and communication cost analysis under LDP and SDP.

**Strengths:**

1. The paper identifies a significant gap in the existing literature on federated linear bandits.
2. The discussion of related works and the comparison of theoretical results are detailed and clear.
3. The algorithm design appears to be reasonable, and while I haven't reviewed the proof in detail, it seems that the theoretical results align with the algorithm's design.

**Weaknesses:**

I don't find any obvious weakness of the paper.

**Questions:**

It appears that the algorithm design in this paper heavily relies on a binary-tree-based mechanism, which can only be applied to linear setting. Is it possible to generalize the algorithm or the analysis to nonlinear models, such as Generalized Linear Models (GLM)?

---

> ### Author Response · Authors · 2023-11-13
> **Response**
>
> Thanks for your positive evaluation of our paper! Regarding your sharp comment on the binary-tree-based algorithm, here is our clarification.
>
> The key behind the binary-tree-based mechanism is the summation structure in parmater estimates. For linear models, the least-square estimate can be written in closed-form using two summations, one over bias vectors and another over covariance matrices. However, for GLM, the maximum likelihood estimate doesn't have a closed-form based on these summations. As a result, one needs to resort to other privacy mechanisms (e.g., objective perturbation), see [R1].
>
> [R1] Chen, Xi, David Simchi-Levi, and Yining Wang. "Privacy-preserving dynamic personalized pricing with demand learning." Management Science 68, no. 7 (2022): 4878-4898.

---

### Official Review · Reviewer_SUqu · 2023-10-31

**Soundness:** 3 good
**Presentation:** 3 good
**Contribution:** 2 fair
**Rating:** 6
**Confidence:** 4

**Summary:**

This work studies the problem of differentially private federated linear contextual bandits. Especially, it first identifies the potential privacy leaking from the adaptive communication strategy adopted in previous works, and the incorrect regret bound. To resolve these issues, this work proposes the Private-FedLinUCB framework, which can flexibly enable both silo-level local DP and shuffle DP. Especially, the shuffle DP guarantee is achieved via two different approaches. Theoretical analyses demonstrated the provable efficiency of the proposed framework. In particular, under SDP, the centralized performance can be approached.

**Strengths:**

- This work identifies the existing issues in the previous study on differentially private federated contextual bandits, which I believe is valuable. Especially, since adaptive communication schemes are widely adopted in studies of federated contextual bandits, it is important to highlight its potential risk of privacy leaking.

- This work nicely combines DP with federated contextual bandits, where the techniques, especially, two types of approaches to obtain shuffle DP, may be of interest for future studies in this line.

- The overall presentation is satisfactory and the study is thorough and complete. Theoretical results are sound based on my understanding.

**Weaknesses:**

- The discussion from federated LDP to SDP is a bit unsmooth in my mind. From the reading, it seems that the authors cannot remove the additional $M^{1/4}$ gap from centralized performance and then the focus is turned to a slightly weaker DP notion of SDP (instead of studying whether the gap can be closed in LDP). I would suggest the authors first justify both DP notions in federated contextual bandits (especially SDP), and then state their corresponding results.

- Related to the first point, it would be nice to add some discussions on whether the $M^{1/4}$ gap from centralized performance can be closed; otherwise, the significance of the result is hard to measure.

- In terms of the DP techniques, I understand that there are many different choices and this work adopts two certain ones. It would be nice to clarify whether the adopted ones are necessary or if there are other feasible choices.

- The fixed batch size selection potentially can be improved. Although performing adaptive communication protocol is no longer feasible, it might still be a good choice to have the batch length exponentially growing (as in many low-switching bandit studies) instead of being a fixed one.

- A recent work [R1] studies a different kind of DP notion in federated contextual bandits. It would be nice to include and discuss it.

[R1] Huang et al. ICML 2023 "Federated Linear Contextual Bandits with User-level Differential Privacy".

**Questions:**

I would love to hear the authors' opinions on my concerns in the weakness part. If I missed or misunderstood anything, please feel free to let me know.

---

> ### Author Response · Authors · 2023-11-13
> **Response -- Part I**
>
> We first thank the reviewer for providing sharp comments and great suggestions. Please find our clarifications below.
>
> **Close the gap under silo-level LDP:** Thanks for the keen comment. It turns out that the first step to obtain a lower bound for Federated LCBs under silo-level LDP is to establish a tight characterization of regret for single-agent LCBs under central JDP.
>
> To see this, one can use a similar reduction as in [Lowy & Razaviyayn' 2021], where the authors derive a lower bound in the supervised learning setting under silo-level LDP. Specifically, for any silo-level LDP algorithm $\mathcal{A}$ with privacy guarantee $\epsilon$, one can first "virtually" shuffle all $MT$ user sequences and then apply $\mathcal{A}$, leading to a shuffled version $\mathcal{A}_s$.
> The shuffled version algorithm $\mathcal{A}_s$ has an SDP privacy guarantee of roughly $\epsilon/\sqrt{M}$. Since SDP implies central JDP in LCBs, one can conclude that $\mathcal{A}_s$ has a lower bound of $L_c(\epsilon/\sqrt{M})$, where  $L_c(\epsilon)$
> denotes the regret lower bound for LCBs under central JDP with privacy $\epsilon'$. Since $\mathcal{A}$ and $\mathcal{A}'$ have same regret performance, this yields a regret lower bound $L_c(\epsilon/\sqrt{M})$ for $\mathcal{A}$ under silo-level LDP.
>
> To our best knowledge, the existing lower bound for LCBs under central JDP is $\Omega(\sqrt{T} + 1/\epsilon')$ [R0], which implies a lower bound $L_c(\epsilon')=\Omega(\sqrt{MT} + 1/\epsilon')$ in the centralized setting (super-single agent). By the above argument, setting $\epsilon'=\epsilon/\sqrt{M}$, this implies a lower bound $\Omega(\sqrt{MT} + \sqrt{M}/\epsilon)$ under silo-level LDP, whereas our upper bound is $O(M^{3/4}\sqrt{T/\epsilon})$. That is, privacy cost in lower bound is only additive $\sqrt{M}/\epsilon$, whereas in our upper bound is a multiplicative $M^{1/4}/\sqrt{\epsilon}$. It is unclear to us which one of these is loose. Hence, whether the regret gap can be closed under silo-level LDP without resorting to SDP remains an open question.
>
>  Now, if one can prove a lower bound $\Omega(\sqrt{T/\epsilon'})$ for single-agent LCBs under central JDP, then it would yield a lower bound $L_c(\epsilon') =\Omega(\sqrt{MT/\epsilon'})$ in the centralized setting, which would further imply a lower bound of $\Omega(M^{3/4}\sqrt{T/\epsilon})$ under silo-level LDP, and would close the regret gap.
>
>  We have included this discussion in Appendix G.1. We believe our findings in this paper (e.g., identifying existing gaps and establishing new upper bounds) would motivate these interesting open questions.
>
> [R0] Jiahao He, Jiheng Zhang, and Rachel Zhang. A reduction from linear contextual bandit lower bounds
> to estimation lower bound, ICML'22
>
> **Other DP techniques:** One can indeed have the flexibility of choosing other DP techniques in our privacy protocol. In fact, this is the beauty of our proposed protocol. For example, one may replace our Vallina tree-based algorithm with other advanced techniques to improve constant factors in privacy parameters, e.g., a low-variance tree-based algorithm [R1] or some online matrix mechanisms [R2, R3]. For silo-level LDP, instead of Gaussian noise, one can use Wishart Noise as in [Sharif&Sheffet'18]. For SDP, instead of vector sum protocol, one can use some advanced SDP protocols, e.g., [R4, R5]
>
> [R1] James Honaker. Efficient use of differentially private binary trees. Theory and Practice of
> Differential Privacy (TPDP 2015), London, UK, 2015
>
> [R2] Denisov, S., McMahan, H. B., Rush, J., Smith, A., & Guha Thakurta, A. (2022). Improved differential privacy for sgd via optimal private linear operators on adaptive streams. Advances in Neural Information Processing Systems, 35, 5910-5924.
>
> [R3] Fichtenberger, H., Henzinger, M., & Upadhyay, J. (2022). Constant matters: Fine-grained Complexity of Differentially Private Continual Observation. arXiv preprint arXiv:2202.11205.
>
>
> [R4] Ghazi, B., Kumar, R., Manurangsi, P., and Pagh, R. Private
> counting from anonymous messages: Near-optimal accuracy with vanishing communication overhead. In International Conference on Machine Learning, pp. 3505–3514.
> PMLR, 2020.
>
> [R5] Balle, B., Bell, J., Gascon, A., and Nissim, K. Private summation in the multi-message shuffle model. In Proceedings of the 2020 ACM SIGSAC Conference on Computer
> and Communications Security, pp. 657–676, 2020

---

> ### Author Response · Authors · 2023-11-13
> **Response -- Part II**
>
> **Exponentially growing batch:** Thanks for the comment. In contrast to standard MABs and linear bandits with stochastic contexts, it is not clear to us how to use exponentially growing batch schedule to derive meaningful regret bound in the setting of linear bandit with *adversarial contexts* and infinite actions (which is the setting of this paper). The only method we are aware of is the *rarely switching principle* (e.g. determinant trick) in [R1], which is the core idea behind adaptive communication in linear bandits, but not feasible in our case due to potential risk of privacy leakage. In fact, [R2] lists an open problem regarding the determinant trick under DP, even in the single-agent setting.
>
> [R1] Abbasi-Yadkori, Y., Pál, D., & Szepesvári, C. "Improved algorithms for linear stochastic bandits". NeurIPS'11.
>
> [R2] Chowdhury, Sayak Ray, and Xingyu Zhou. "Shuffle private linear contextual bandits." ICML 2022
>
> **Discuss on recent work on user-level LCBs:** Thanks for pointing out this nice recent related work. We have included it in the main paper (highlighted in brown color on page 3). We have also provided the following discussions in Appendix A.
>
> “Recently, Huang et al. (2023) took the pioneering step to study user-level privacy for federated LCBs,
> establishing both regret upper bounds and lower bounds. In contrast to our item-level DP (e.g.,
> silo-level LDP), user-level DP in Huang et al. (2023) roughly requires that even replacing the whole
> local history at any agent, the central server’s broadcast message should be close across the whole
> learning period. This notion is more likely to be preferred in cross-device FL settings where the
> protection target is the device (agent). In addition to this, there are several key differences compared to
> our work. First, they deal with linear bandits with stochastic contexts under additional
> distribution coverage assumptions (rather than the arbitrary adversary contexts in our case). In fact, it has
> been shown by Huang et al. (2023) that some assumption on the context distribution is necessary
> for a sublinear regret under user-level DP. Second, due to this stochastic context and some coverage
> conditions on contexts, an exponentially growing batch schedule can be applied in their case. In
> contrast, under the adversary context case, it is unclear to us how to apply the same technique to
> derive a sublinear regret.”
>
> **On smooth transition to SDP:** Thanks for the suggestion. We have updated the transition from silo-level LDP to SDP, highlighted in brown color on Page 3.

---

> > ### Comment · Reviewer_SUqu · 2023-11-23
> >
> > Thank you for the responses! They have largely resolved my concerns, especially regarding the batch size and the DP mechanism. The discussions on lower bounds are also inspiring. I will keep my score for now and discuss with other reviewers.

---

### Meta-Review · Area_Chair_75gb · 2023-12-05

**Metareview:**

This paper fills an important gap in the bandit literature. Reviewers were unanimous about the contributions made, the meticulous efforts in exposition (including the discussion on shuffle DP), and the fact that it identifies an error in a previous paper (Dubey & Pentland 2020), setting the record right. It is clear that the machine learning community would benefit from this paper.

**Justification For Why Not Higher Score:**

The paper, while being a valuable contribution to the community, can't be considered groundbreaking.

**Justification For Why Not Lower Score:**

The contributions and exposition are clear. The paper fills important gaps in the bandit literature and may inspire many future works in this direction.

---

### Decision · Program_Chairs · 2024-01-16

Accept (poster)